# A single-cell analysis of breast cancer cell lines to study tumour heterogeneity and drug response

G. Gambardella [1,2,4], G. Viscido[1,2,4], B. Tumaini[1], A. Isacchi[3], R. Bosotti [3] & D. di Bernardo [1,2 ✉]

Cancer cells within a tumour have heterogeneous phenotypes and exhibit dynamic plasticity. How to evaluate such heterogeneity and its impact on outcome and drug response is still unclear. Here, we transcriptionally profile 35,276 individual cells from 32 breast cancer cell lines to yield a single cell atlas. We find high degree of heterogeneity in the expression of biomarkers. We then train a deconvolution algorithm on the atlas to determine cell line composition from bulk gene expression profiles of tumour biopsies, thus enabling cell line-based patient stratification. Finally, we link results from large-scale in vitro drug screening in cell lines to the single cell data to computationally predict drug responses starting from single-cell profiles. We find that transcriptional heterogeneity enables cells with differential drug sensitivity to co-exist in the same population. Our work provides a framework to determine tumour heterogeneity in terms of cell line composition and drug response.

[1] Telethon Institute of Genetics and Medicine, Naples, Italy. [2] University of Naples Federico II, Department of Chemical, Materials and Industrial Engineering, Naples, Italy. [3] NMSsrl, Nerviano Medical Sciences, 20014 Nerviano, Milan, Italy. [4] These authors contributed equally: G. Gambardella, G. Viscido. ✉email: dibernardo@tigem.it

One of the main roadblocks to personalized medicine of cancer is the lack of biomarkers to predict outcome and drug sensitivity from a tumour biopsy. Multigene assays such as MammaPrint[1], Oncotype DX[2,3] and PAM50[4] can classify Breast Cancer (BC) tumour types and risk of relapse[5] but with limited clinical value[5,6]. Genomic and transcriptional biomarkers of drug sensitivity are available only for a restricted number of drugs[7–9]. As a consequence, BC patient stratification is still mainly driven by receptor status and histological grading and subtyping[5], with about twenty percent[10] of patients for which paucity of actionable biomarkers limits personalized therapies. Moreover, even when a targeted treatment option is available, drug resistance may arise[5] partly because of rare drug-tolerant cells characterized by distinct transcriptional or mutational states[11–17].

Determining tumour heterogeneity and its impact on drug response is essential to better stratify patients and aid in the development of personalized therapies. Expression-based biomarkers measured from bulk RNA-sequencing of a tumour biopsy are powerful predictors of drug response in vitro[7,8,18], but average out tumour heterogeneity. Single-cell transcriptomics yields a molecular profile of each cell[19,20]; however, it is still unclear if and how it can inform clinical decision making.

Here, we transcriptionally profile 35,276 individual cells from 32 breast cancer cell lines. We show that despite being simplistic models of tumours, cancer cell lines exhibit themselves heterogeneous phenotypes, and can serve as cell-state "primitives" to deconvolve tumour cell composition from patients' biopsies for patient stratification and prediction of drug response. By linking results from large-scale in vitro drug screening in cell lines to the single-cell data, we devise an algorithm to computationally predict drug responses starting from single-cell profiles. We find that non-genetic transcriptional heterogeneity enables cells with differential drug sensitivity to co-exist even in the same population. Our work provides a framework to characterize intra-tumoral heterogeneity from gene expression profiles in terms of cell-line composition and differential sensitivity to drug treatment.

## Results

**Single-cell transcriptome profiling of breast cancer cell lines.** We performed single-cell RNA-sequencing (scRNA-seq) of 31 breast cancer cell lines, 16 of which from metastatic sites (Supplementary Table 01 and Supplementary Table 02), plus one additional non-cancer cell line (MCF12A[21]) by means of the Drop-seq technology[20]. We chose this set of cell lines as they cover all the major breast cancer tumour subtypes (LuminalA/LuminalB/Her2-enriched/Basal Like) and are extensively used in cancer research and in drug development, while also being fully characterized both at the genomic and (bulk) transcriptomic level, as well as in terms of drug response[7,8,22,23].

Following pre-processing (Methods), we retained a total of 35,276 cells, with an average of 1069 cells per cell line and 3248 genes captured per cell (Supplementary Fig. 01 and Supplementary Table 01).

We next generated an atlas (http://bcatlas.tigem.it) encompassing the 32 cell lines, as shown in Fig. 1A. In the atlas, it is possible to recognize a luminal-supergroup with intermixing of cells from different luminal cell lines and Her2-enriched (Her2+) cell lines, while triple-negative breast cancer (TNBC) cell lines form distinct clusters, thus suggesting that these represent instances of different diseases. We investigated if genomic features could explain the formation of such clusters. To this end, we clustered cell lines according to either genomic variants or Copy Number Variations (CNV)[24]. Clustering according to genomic variants, shown in Supplementary Fig. 02A, did not yield any meaningful clustering.

On the contrary, clustering according to CNVs yielded eight distinct clusters, as shown in Supplementary Fig. 02B. We mapped these CNV-based clusters onto to atlas, as shown in Supplementary Fig. 02C, to check whether CNVs can explain some of the features of the single-cell clustering; we found no obvious pattern: for example, the CNV-based cluster 5 (cyan) contains three cell lines (AU565, BT474 and T47D) with similar CNVs; however, the Her2 + AU565 cell line forms a distinct cluster in the single-cell atlas, while the luminal BT474 and T47D cell lines belong to the luminal-supergroup; similarly the CNV-based cluster 4 (blue) contains three cell lines (CAL51, BT549 and HS578T) that, however, form distinct clusters in the single-cell atlas.

Single-cell expression of clinically relevant biomarkers (Fig. 1B, C) including oestrogen receptor 1 (ESR1), progesterone receptor (PGR), Erb-B2 Receptor Tyrosine Kinase 2 (ERBB2 a.k.a. HER2) and the epithelial growth factor receptor (EGFR) across the different cell lines are in agreement with their reported status[21,25,26].

To gain further insights into each cancer cell line, we analyzed the expression of 48 literature-based biomarkers of clinical relevance[27], as reported in Fig. 1D and Supplementary Fig. 03. Luminal cell lines highly express luminal epithelium genes, but neither basal epithelial nor stromal markers; on the contrary, triple-negative BC cell lines show a basal-like phenotype (9 out of 15 as quantified in Supplementary Table 03) with the expression of at least one of keratin 5, 14 or 17[28,29], with triple-negative subtype B cell lines also expressing vimentin (VIM) and Collagen Type VI Alpha Chains (COL6A1, COL6A2, COL6A3)[21]. Interestingly, two out of five HER2+ cell lines (JIMT1 and HCC1954) are close to triple-negative cell lines in the atlas and express keratin 5 (KRT5) (Fig. 1A, D), which has been linked to poor prognosis and trastuzumab resistance[30]. Indeed, both cell lines are resistant to anti-HER2 treatments[31]. Finally, the non-tumorigenic MCF12A cell line lacks expression of ESR1, PGR and HER2 and displays a basal-like phenotype characterized by the expression of all basal-like marker genes including keratin 5, 14, 17 and TP63, in agreement with the literature[32].

Overall, these results show that single-cell transcriptomics can be successfully used to capture the overall expression of clinically relevant markers.

**The BC single-cell atlas identifies clinically relevant transcriptional signatures.** By clustering the 35,276 single cells in the atlas, we identified 22 clusters, as shown in Fig. 1E. Within the luminal supergroup, cells did not cluster according to their cell line of origin, indeed four out of the five luminal clusters contain cells from distinct cell lines (Fig. 1F and Supplementary Fig. 04). On the contrary, triple-negative cell lines tend to cluster according to their cell line of origin, with each cluster containing mostly cells from the same cell line (Fig. 1F and Supplementary Fig. 04).

We identified genes differentially expressed between cells in the same cluster against all the remaining cells in the atlas. We then selected one gene for each cluster (i.e. the most differentially expressed) for a total of 22 cluster-derived biomarkers (Fig. 1G, H and Supplementary Fig. 05). Neither ESR1 nor ERRB2 were part of this set. Literature mining confirmed the significance of some of these genes: biomarkers from the luminal supergroup clusters (Fig. 1G) were associated with cancer progression (BCAS3[33,34] cluster 2), dissemination (SCGB2A2[35,36] cluster 6), proliferation (DRAIC[37,38] cluster 1), migration and invasion (CLCA2[39,40] cluster 8 and PIP[41] cluster 18). Interestingly, whereas DRAIC is correlated with poorer survival of luminal BC patients[38], both CLCA2 and PIP are significantly associated with a favourable prognosis[39,40,42,43].

To examine the clinical relevance of these 22 biomarkers, we analyzed their expression across 937 breast cancer patients from

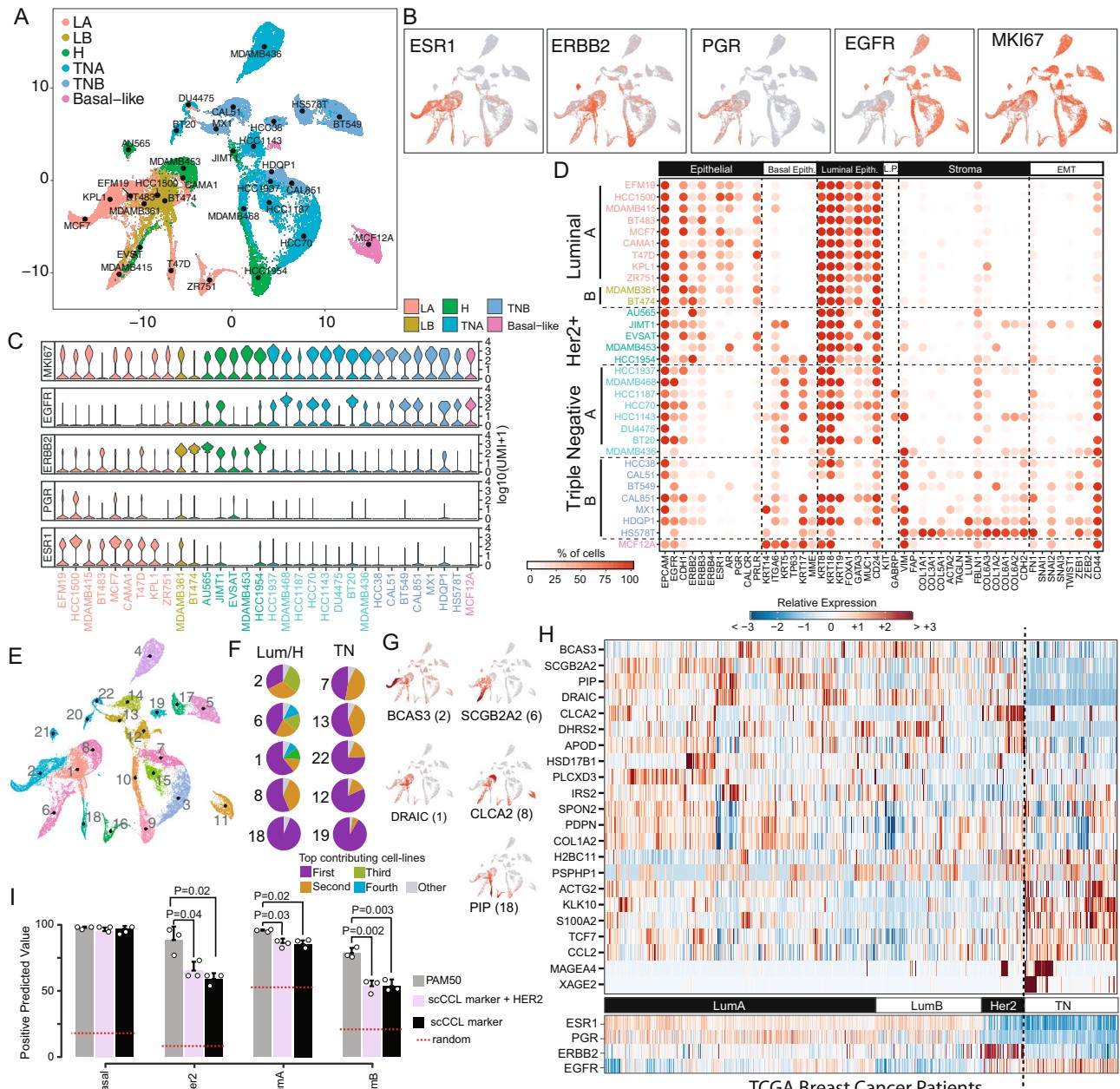

**Fig. 1 The breast cancer single-cell atlas. A** Representation of single-cell expression profiles of 35,276 cells from 32 cell lines color-coded according to cancer subtype (LA luminal A, LB luminal B, H Her2-enriched, TNA triple-negative type A, TNB triple-negative type B). **B** Expression levels of the indicated genes in the atlas, with red indicating expression, together with their **C** distribution within the cell lines, shown as a violin plot. **D** Dotplot of literature-based biomarker genes along the columns, for each of the 32 sequenced cell lines along the rows. Biomarker genes are grouped by type (Basal Epith. basal epithelial, Luminal Epith. luminal epithelial, L.P. luminal progenitor, EMT Epithelial to Mesenchymal Transition). **E** Graphical representation of 35,276 cells color-coded according to their cluster of origin. Clusters are numbered from 1 to 22. **F** For the indicated cluster, the corresponding pie chart represents the cluster composition in terms of cell lines. Cell lines in the same pie chart are distinguished by colour. Only the top 10 most heterogenous clusters are shown. In grey cell lines in the cluster contributing less than 5%, while the other colours represent a specific cell line. For example, Cluster 2 is the most heterogeneous cluster mainly composed of 3 cell line while cluster 19 is the most homogenous since in its mainly composed by the cells coming from one cell line. **G** Expression levels in the atlas of the five luminal biomarkers identified as the most differentially expressed in each of the five luminal clusters (1, 2, 6, 8 and 18). **H** Expression of 22 atlas-derived biomarkers in the biopsies of 937 breast cancer patient from TCGA. **I** Accuracy in classifying tumour subtype for 937 patients from TCGA by using either the PAM50 gene signature or the 22 cluster-derived biomarker genes (scCCL) alone or augmented with HER2 gene (scCCL + HER2). Two-sided *t*-test is used to compare the performance of the different signatures. Source data are provided in a Source data file.

the Cancer Genome Atlas (TCGA) collection encompassing all four BC types. As shown in Fig. 1H, and quantified in Supplementary Table 04, there is a significant difference in the expression of the 22 cluster-derived biomarkers across Luminal A, Luminal B, Her2+ and Triple Negative patients. Moreover, it is possible to distinguish subtypes within each category, which may lead to additional diagnostic/prognostic biomarkers (Fig. 1H). For example, two of the biomarkers (MAGE4 and

XAGE4) are highly expressed only in a subset of triple-negative breast cancer patients and of HER2 + /ER− patients (Fig. 1H); interestingly, one of the two (MAGE4) has been previously reported in the literature as overexpressed in such patients by proteomic profiling[44]. The second subset of triple-negative patients is characterized by actin gamma 2 expression (ACTG2), which has been previously linked in BC to cell proliferation[45] and platinum-based chemotherapy sensitivity[46–49]. We observed that two triple-negative cell lines in the atlas (HS578T and MX1) showed considerably higher expression of ACTG2 than all the other cells in the atlas (Supplementary Fig. 06A, B). To confirm the link with cis-platin sensitivity, we treated both cell lines with cis-platin and measured cell viability at 72 h at different dosages, as shown in Supplementary Fig. 06C and Supplementary Table 05. These results confirm cis-platin sensitivity of both cell lines, albeit higher in HS578T cells than in MX1 cells.

Finally, to further confirm the clinical relevance of these 22 cluster-derived biomarker genes, we compared their performance in correctly classifying BC subtypes from bulk RNA-seq data of TGCA patients against the clinically-approved PAM50 gene signature (50 genes)[4]. As shown in Fig. 1I, classification performances were better than random for all the four subtypes but comparable with the PAM50 only for the basal subtype, whereas HER2-overexpressing cancers had the worst performance. As expected, when adding *ERBB2* to the list of 22 cluster-based biomarkers, the classification of this subtype improved (Fig. 1I). It is important to observe that, unlike the PAM50, the 22 biomarkers were automatically derived from the single-cell atlas without using any prior knowledge of breast cancer subtypes.

Altogether, these analyses confirm that the single-cell BC cell-line atlas can be used for automatic identification of clinically relevant genes that can be useful for patient stratification and tumour type classification.

**The BC atlas as a reference for automated cancer diagnosis.** The BC atlas can be used as a reference against which to compare single-cell transcriptomics data from a patient's tissue biopsy and to perform cancer subtype classification and assessment of tumour heterogeneity. To this end, we developed an algorithm able to map single-cell transcriptional profiles from a patient onto the BC atlas and to assign a specific cell line to each of the patient's cells (Methods and Supplementary Fig. 07). We tested the ability of the algorithm in correctly mapping the very cells in the atlas from their single-cell transcriptional profile by first dividing single-cell transcriptional profiles in the atlas in a training set (75% of the cells in each cell line), and a test set (25% of cells in each cell line). As shown in Supplementary Fig. 08, the accuracy of the mapping algorithm on the test set was greater than 75% of correctly mapped cells for most of the cell lines (28 out of 32). We then mapped onto the BC atlas the publicly available[50] single-cell transcriptional profiles obtained from five triple-negative breast cancer patients enrolled in a clinical trial for neoadjuvant chemotherapy treatment with a pathological evaluation of haematoxylin and eosin-stained tissue sections, immunohistochemistry analysis of oestrogen receptor (<1%) and progesterone receptor (<1%) and fluorescence in situ hybridization analysis of HER2 amplification (ratio of HER2 to CEP-17 < 2.2). As shown in Fig. 2A, B, most patients' cells mapped to the triple-negative clusters as expected, except for the TNBC5 patient's sample, for which most cells mapped to the luminal supergroup. Interestingly, TNBC5 was the only patient highly expressing both the androgen receptor AR and the transcription factor FOXA1 (Supplementary Fig. 09). Co-expression of these two genes has been reported in the literature to occur in about 15% of triple-negative breast patients, and it is considered a

distinct class of basal-like tumour inducing a luminal-like gene signature[51,52]. This observation suggests that patient TNBC5 cells were mapped to luminal cell lines, as the algorithm found those cell lines to be the most similar in the atlas. To further investigate TNBC5 unusual expression profile, we applied the PAM50 signature to the pseudo-bulk expression profiles of the five TNBC patients. Pseudo-bulk refers to the use of single-cell expression profiles to compute the average gene expression and thus simulate a bulk gene expression measurement. The results of the PAM50 classification are reported in Supplementary Table 06 and show that whereas patients TNBC1, 2, 3 and 4 were correctly classified as basal-like with about 99% probability, on the contrary TNBC5 has only a 4% probability of being basal-like, compared to a 47% probability of being HER2-enriched, and 48% probability of being luminal, in agreement with our mapping algorithm predictions and further confirming the peculiarity of this patient. These results demonstrate that heterogeneity varies across patients but is present in all the samples, as no patient's biopsy mapped to a single cell line. Moreover, information on the drug sensitivity of the individual cell lines composing the tumour may prove useful in guiding therapeutic choices.

We next tested the algorithm on publicly available[53] spatial transcriptomics dataset obtained from tissue biopsies of two patients, one diagnosed with ESR1$^+$/ERBB2$^+$ lobular oestrogen-positive carcinoma (Fig. 2C and Supplementary Fig. 10A) and the other with ESR1$^+$/ERBB2$^+$ ductal carcinoma (Supplementary Fig. 10B,C). The publicly available dataset includes 3808 transcriptional profiles for patient 1 (Fig. 2C) and 3615 profiles for patient 2 (Supplementary Fig. 10B,C), each obtained from a different tissue "tile" of size $50 \times 50 \times 50$ um. The IHC and HER2 FISH data used for the diagnosis were not publicly available. The algorithm projected each of the spatial tiles onto the BC atlas and assigned a cell line to each tile. We coloured the tiles according either to the mapped cell line or to the BC subtype of the mapped cell line (Fig. 2C) to yield an automatic cancer subtype classification of tiles. Most of the tiles for both patients were assigned to just two cell lines and correctly classified as luminal (A or B); the remaining 13% of the tiles for patient 1 and 20% for patient 2 were instead classified either as HER2-overexpressing or triple-negative, which could be important information to guide therapeutic choice and to predict the occurrence of drug resistance. Since spatial data do not have a single-cell resolution, each spatial tile could also be itself a mixture of heterogeneous profiles. Thus an alternative approach is to use bioinformatics tools, such as Cell2Location[54], which can be trained on the BC single-cell atlas and used to estimate the cell-line composition of each spatial tile, rather than attempting to assign the entire tile to just one cell line. The results of applying Cell2Location on the tissue biopsies of the two patients are reported in Supplementary Fig. 11 and Supplementary Fig. 12.

As bulk gene expression profiles are more clinically relevant than single-cell gene expression profiles, we next trained a recently published bioinformatics tool named Bisque[55] (Methods) on our single-cell atlas to predict the cell-line composition of a tumour sample. Bisque was originally devised to estimate cell type proportions from bulk RNA-seq data of complex tissues. To test the effectiveness of Bisque in our settings, we first applied it to bulk RNA-seq transcriptomic data of breast cancer cell lines that are publicly available in the CCLE[24] database and that were also present in our atlas (i.e. 29 out of 32 cell lines). We then used Bisque to predict from the bulk gene expression profile of each cell line, its composition using the single-cell transcriptional profiles in the atlas. As shown in Supplementary Fig. 13, for each of the 29 bulk gene expression profiles, Bisque correctly predicted that the largest fraction of cells composing it came from the corresponding cell line in the atlas with a range between 40% and 80%.

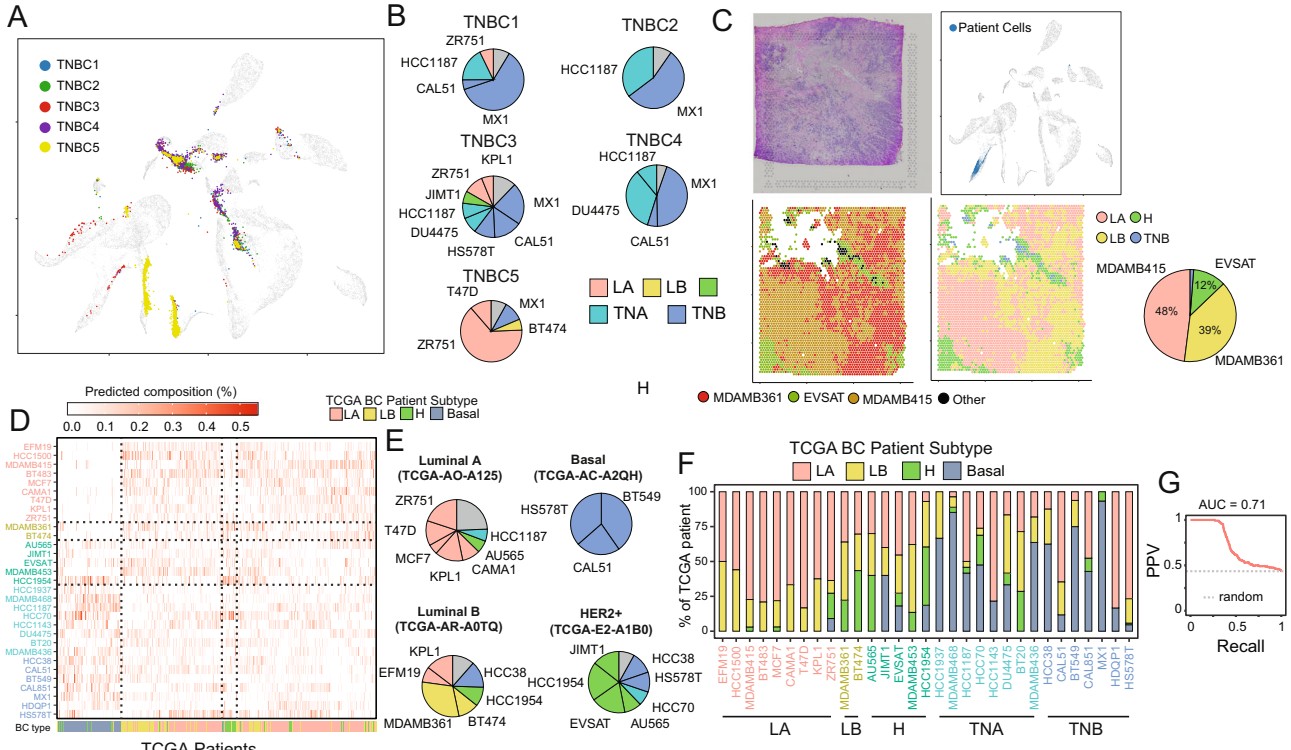

**Fig. 2 Automatic classification of patients' tumour cells. A** Cancer cells from triple-negative breast cancer (TNBC) biopsies of five patients were embedded in the BC atlas by means of the mapping algorithm in order to predict their tumour subtype. **B** For each patient, the pie chart shows cell-line composition obtained by mapping patient's cells onto the atlas. **C** Tissue-slide of an oestrogen-positive breast tumour biopsy sequenced by means of the 10× Genomics Visium spatial transcriptomics (top-left) and the position of the mapped tissue tiles onto the atlas (top-left). Tiles are colour-coded according to the cell line (bottom-left) and to tumour subtype (bottom-right) as predicted by the mapping algorithm. **D** Cell-line composition of 937 BC patients from the TGCA database as estimated by the Bisque algorithm from their bulk RNA-seq data. For ease of interpretation, in the heatmap patients are clustered according to their cell-line composition. The bottom row reports the annotated cancer subtype in TGCA. **E** Predicted cell-line composition by the Bisque algorithm for four representative patients. **F** The distribution of the 937 BC patients across the 32 cell lines. For each cell line, the stacked bars report the percentage of patients of a given cancer subtype assigned by Bisque to that cell line. Since each patient is usually predicted by Bisque to be composed by multiple cell lines, the patient is associated to the cell line making up the largest fraction of the patient's cell-line composition. **G** Performance of Bisque in classifying the tumour subtype of the 937 BC patients in TGCA from bulk gene expression profiles. Since each patient is usually predicted by Bisque to be composed by multiple cell lines, the patient is associated to the tumour type of the cell line making up the largest fraction of the cell-line composition. (PPV positive predictive value, AUC area under the curve). Source data are provided in a Source data file.

We then applied Bisque to 937 bulk gene expression profiles from breast cancer patients in TGCA whose BC subtypes were annotated, and then assigned to each patient the corresponding cell-line composition as shown in Fig. 2D, E. Reassuringly, patients diagnosed with a specific breast cancer subtype tend to have a tumour cell-line composition consisting of cell lines of the same subtype. We quantified this observation in Fig. 2F and observed some interesting exceptions: JIMT-1 is a HER2-overexpressing cell line with an amplified ERBB2 locus, but for no HER2$^+$ patient Bisque predicted the JIMT-1 cell line as the one making up the largest fraction of the patient's cell-line composition. Interestingly, JIMT-1 cells are resistant to anti-HER2 treatments[56]; another example is the HS578T cell line, which is reported to be triple-negative; however, the majority of patients who map to it are luminal; interestingly, this cell line has been reported to be sensitive to fulvestrant[7,8], an anti-ESR1 drug. We finally quantified the performance of the Bisque algorithm trained on the single-cell atlas in correctly classifying the tumour subtype of the 937 TGCA patients from bulk RNA-seq. To this end, we assigned to each patient the tumour subtype of the cell line making up the largest fraction of the patient's cell-line composition. Figure 2G reports the classification performance in terms of precision-recall curve, achieving an Area Under the

Curve of 0.71. Altogether, these results show that the BC single-cell atlas can be used to automatically assign cell-line composition and cancer subtypes both from single-cell expression profiles and bulk gene expression profiles.

**Clinically relevant biomarkers exhibit heterogenous and dynamic expression in BC cell lines.** Clinically relevant receptors are heterogeneously expressed across cells belonging to the same cell line, as assessed by computing the percentage of cells in a cell line expressing the receptor as in Fig. 3A. Consider the seven Luminal B and HER2$^+$ cell lines present in the BC atlas, which by definition overexpress HER2: whereas more than 90% of cells in AU565, BT574 and HCC1954 cell lines express ERBB2, in the remaining four cell lines ERBB2 expression ranged from 31% of EVSAT cells to 46% of JIMT1 cells and up to 64% of MDA-MB-361 cells. This happens despite both JIMT1 and MDA-MB-361 harbouring a copy number gain of the locus containing the ERBB2 gene[57]. We first excluded the possibility that these results were artifacts of single-cell RNA-sequencing technology by showing that estimated BC receptor heterogeneity is not correlated to sequencing depth (Supplementary Fig. 14), and by a simulation approach assuming a Poisson sampling of sequencing data[58,59] as reported in Supplementary Table 07 (Methods). More

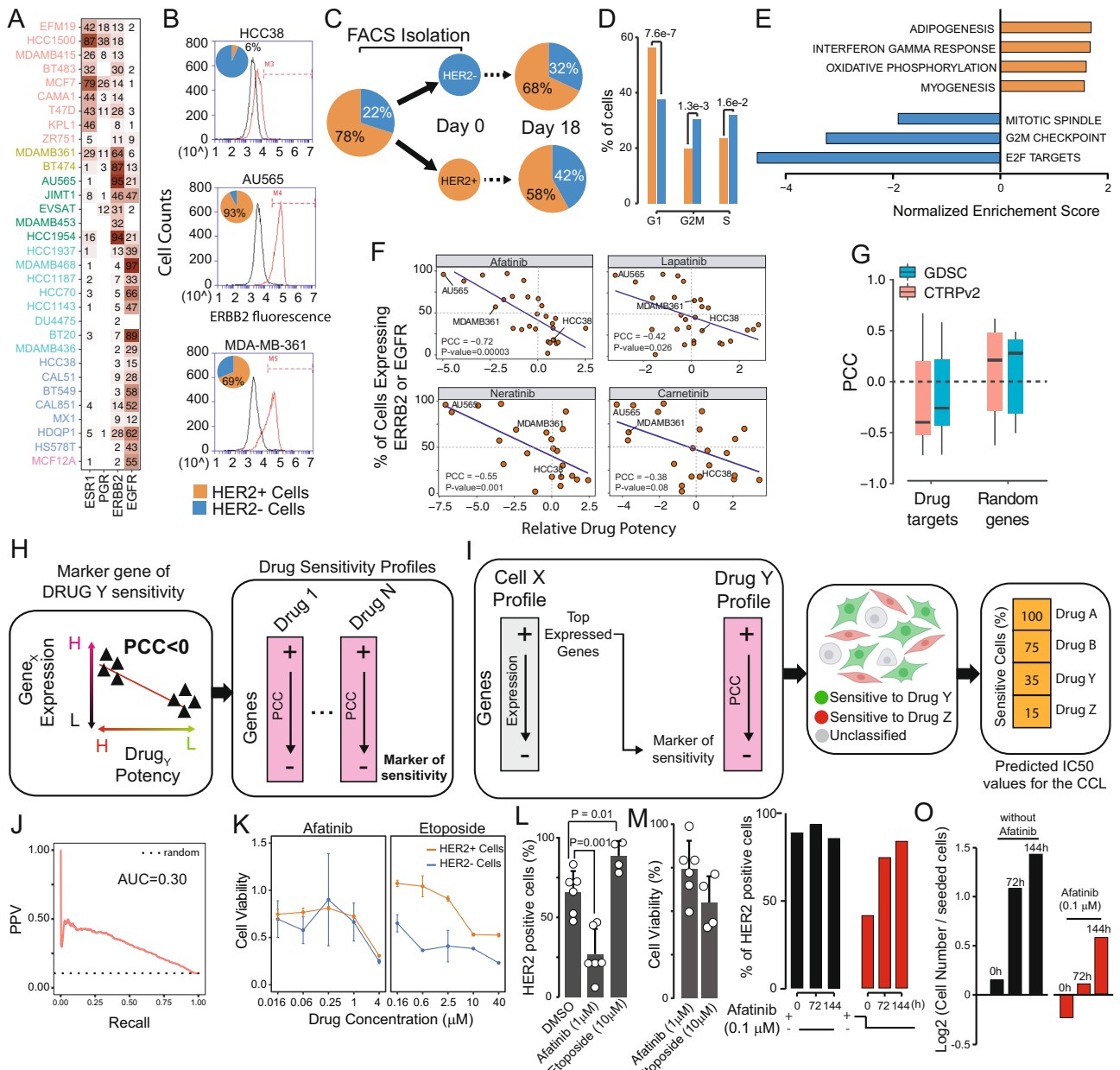

**Fig. 3 Transcriptional heterogeneity in breast cancer cell lines and its impact on drug response. A** Percentage of cells expressing the indicated genes in each of the 32 cell lines. **B** Fluorescence cytometry of HCC38, MDA-MB-361 and AU565 cell lines stained with a fluorescent antibody against HER2. **C** Expression of HER2 protein in MDA-MB-361 cells is dynamic and re-established in less than 3 weeks. **D** Cell cycle phase for the HER2$^+$ and HER2$^-$ subpopulations of MDA-MB-361 cells. p-value refers to the Fisher's exact test. **E** Enriched pathways (GSEA, FDR < 10%) across differentially expressed genes between the HER2$^+$ (orange) and HER2$^-$ (blue) MDA-MB-361 cells. **F** Gene expression versus drug potency for four anti-HER2 drugs. Each dot corresponds to a cell line with percentage of cells expressing ERBB2 or EGFR [y-axis] versus the experimental drug potency[8] as Area Under the Curve (AUC) [x-axis]. PCC (Pearson correlation coefficient) and its p-value are also shown. **G** PCC values computed as in F for 66 drugs for which the cognate drug targets is known. The PCC distribution when choosing a random gene is also shown. Boxplots containing PCC distribution between a random gene and drug $n = 1000$, while $n = 66$ for boxplot containing PCC distribution between a drug and its cognate target gene. **H** Bioinformatics pipeline for the identification of drug sensitivity biomarkers for 450 drugs. **I** The top 250 most expressed genes in a single cell are used as input for a GSEA against the ranked list of genes correlated with drug potency for each one of the 450 drugs to predict its drug sensitivity. **J** Performance of DREEP in predicting drug sensitivity of 32 cell lines in the atlas to 450 drugs in terms of PPV (Positive Predicted Value) versus Recall. **K** Dose-response curve in terms of cell viability following treatment with either afatinib or etoposide at the indicated concentrations on sorted MDA-MB-361 cells (triplicate experiment). **L** Percentage of HER2$^+$ cells in MDA-MB-361 after 72 h treatment with either afatinib or etoposide. (two-sided t-test), and **M** cell viability. **N** Percentage of HER2$^+$ cells in MDA-MB-361 cell line at the indicated time-points either following 48 h of afatinib pre-treatment (red bars) or without any afatinib pre-treatment (black bars) and **O** the relative number of cells rescaled for the number of cells at the beginning of the experiment. Source data are provided in a Source data file. For **K**, **L** $n = 3$.

specifically, we computed for each cell line, the expected proportion of zero counts across cells for each of the four clinical biomarkers in Fig. 3A. We then tested whether the actual zero proportion was higher than expected under the Poisson model, as zero inflation indicates the presence of cell heterogeneity[6]. We thus found that heterogeneity in the expression of the clinical biomarkers is significant (p-values < 0.05) for at least one of the four biomarkers in all the cell lines but two (ZR751 and BT549). Moreover, for the MDA-MB-361 cell lines, ESR1, PGR and ERBB2 were all found to be significantly heterogeneously expressed (Supplementary Table 07). We also assessed HER2 protein levels by flow cytometry in three representative cell lines: AU565 (high HER2 expression), MDA-MB-361 (heterogeneous HER2 expression) and HCC38 cell lines (low HER2 expression). As shown in Fig. 3B, single-cell transcriptional data agree with the cytometric analyses; however, the origin of this heterogeneity is unclear. To exclude hereditable genetic differences as a source of heterogeneity, we sorted MDA-MB-361 cells into HER2+ and HER2− subpopulations (Methods) and checked whether these homogenous subpopulations were stable over time, or rather spontaneously gave rise to heterogeneous populations. As shown in Fig. 3C, after 18 days in culture, both subpopulations re-established the original heterogeneity, demonstrating that HER2 expression in these cells is dynamic and driven by a yet undiscovered mechanism.

Interestingly, HER2+ circulating tumour cells (CTCs) isolated from an ER+/HER2− breast cancer patient were previously shown to spontaneously interconvert from HER2− and HER2+, with cells harbouring a phenotype producing daughters of the opposite one[60]. To check if the cell cycle phase could explain the observed heterogeneity in the MDA-MB-361 cell line, we computationally predicted (Methods) the cell cycle phase of each cell in both the HER2− and HER2+ subpopulations from single-cell transcriptomics data[61]. As shown in Fig. 3D, a higher proportion of HER2− cells was predicted to be in the S and G2/M phases when compared to HER2 + cells, with a concomitant lower proportion in the G1 phase. This result is consistent with previous observations that report cell cycle arrest in the G2/M phase following HER2 inhibition[62].

We next set to identify biological processes differing between the two subpopulations by computing differentially expressed genes (DEGs) from the single-cell transcriptional profiles of HER2+ cells against HER2− cells (Supplementary Data 01). Gene Set Enrichment Analyses (GSEA)[63] against the ranked list of DEGs, reported in Fig. 3E, revealed just seven significantly enriched pathways (FDR < 10%): four of which were upregulated in HER2+ cells, but downregulated in HER2− cells, and included adipogenesis, myogenesis and OXPHOS, all indicative of epithelial-to-mesenchymal transition (EMT) engagement, which has been previously observed in HER2+ cells[64–66]; the remaining three pathways were upregulated in HER2− cells and related to cell cycle and specifically to G2/M phase, in agreement with our previous analysis, suggesting that cell cycle may play a role in HER2 expression in this cell line.

These results show that heterogeneity in the expression of clinically relevant biomarkers is present even in cancer cell lines and that it can also be dynamic and of a non-genetic nature.

**Heterogeneity in gene expression affects drug response**. To investigate the role of heterogeneity in gene expression on drug response, we collected large-scale in vitro drug screening data[7,8] reporting the effect of 450 drugs on 658 cancer cell lines from solid tumours. As shown in Fig. 3F, Supplementary Fig. 15 and Supplementary Table 08, the sensitivity of the BC cell lines to HER2 inhibitors was significantly correlated with the percentage

of cells in the cell line expressing ERBB2. This result holds true even when using bulk gene expression of the cell lines (available in the CTRPv2 dataset from the Cancer Cell Line Encyclopaedia —CCLE[24]), in place of the percentage of cells (Supplementary Fig. 16). Interestingly, at the single-cell level receptor expression is substantially the same across cells expressing it, irrespective of the cell line they belong to (Supplementary Fig. 17), except for cell lines harbouring CNVs of the ERBB2 locus. Furthermore, by analyzing all the drugs in the CCLE[24] database for which the cognate target is known, we found that the correlation between drug target expression and drug sensitivity holds true also for 66 drugs out of 302 targeted drugs across CTRPv2 and GDSC datasets (Fig. 3G and Supplementary Data 02). These results suggest that variability in gene expression within cells of the same tumour caused by cellular heterogeneity may cause some cells to respond poorly to the drug treatment.

Starting from these observations, we developed DREEP (DRug Estimation from single-cell Expression Profiles), a bioinformatics tool that, starting from single-cell transcriptional profiles, allows to predict drug response at the single-cell level. To this end, we first detected expression-based biomarkers of drug sensitivity for 450 drugs[8], as schematized in Fig. 3H, I (Methods). Briefly, we crossed data from the CTRPv2 dataset from the CCLE[24] on the response to 450 drugs across 658 cancer cell lines from solid tumours with the cell line gene expression profiles from bulk RNA-seq. In the CCLE, drug potency is evaluated as the inverse of the Area Under the Curve (AUC) of the dose-response graph, with low values of the AUC indicating drug sensitivity, while high values implying drug resistance (Fig. 3H). For each gene and for each drug, we computed the correlation between the expression of the gene across the 658 cell lines with the drug potency in the same cell lines. Hence, genes positively correlated with the AUC are potential markers of resistance, vice-versa, negatively correlated genes are markers of sensitivity (Fig. 3H). In this way, we generated a ranked list of expression-based biomarkers of drug sensitivity and resistance for each of the 450 drugs. We then used these biomarkers to predict drug sensitivity at the single-cell level for the 32 cell lines in the atlas, as depicted in Fig. 3I. To this end, for each cell in the atlas, we compared the 250 genes most expressed by the cell to the ranked list of biomarkers for each one of 450 drugs by means of Gene Set Enrichment Analysis (GSEA)[63], resulting in 450 Enrichment Scores (ES) with corresponding p-values. Finally, the cell was deemed to be sensitive to the drug associated with the most negative ES. If no significant and negative ES score was found then the cell was annotated as unclassified. To convert predictions from the single-cell level to the cell-line level, we chose the drug that was predicted to work in the largest fraction of cells in the cell line. We tested DREEP's performance in predicting the drug sensitivity of the 32 cell lines in the atlas starting from their single-cell transcriptomics data. We chose two independent "golden standards", one derived from the experimental drug potency data of 450 drugs across 658 cancer cell lines in the CTRPv2 dataset, and the other derived from Genomics of Drug Sensitivity in Cancer (GDSC) study[9], which includes drug potency data measured as Inhibitory Concentration at 50% (IC50) for about 250 small molecules (of which only 86 in common with the CTRPv2 dataset). The overall performance across the 32 cell lines in the atlas is reported for the CTRPv2 golden standard in Fig. 3J and for each of the 450 drugs separately in Supplementary Data 03, while Supplementary Fig. 18 reports the performance for the GDSC golden standard. In all cases DREEP performance was better than random.

To experimentally validate DREEP, we turned to the MDA-MB-361 cell line for which we found the coexistence of two distinct and dynamic cell subpopulations (HER2+ and HER2−).

We applied DREEP to each subpopulation to identify drugs able to selectively inhibit the growth of either the HER2⁻ subpopulation or the HER2⁺ subpopulation: 42 drugs (FDR < 1%, Supplementary Table 04) were predicted to preferentially inhibit the growth of HER2⁻ cells; the most overrepresented class among these drugs was that of inhibitors of DNA topoisomerases (TOP1/TOP2A) (Supplementary Figs. 19, 20) such as Etoposide. Surprisingly, no drug was found to specifically inhibit the growth of HER⁺ cells, whereas 44 drugs (FDR < 1%) were predicted to be equally effective on both subpopulations and unexpectedly included HER2 inhibitors, such as afatinib (Supplementary Data 04).

We selected etoposide and afatinib for further experimental validation. MDA-MB-361 cells were first sorted by FACS into HER2⁺ and HER2⁻ subpopulations and then cell viability was measured following 72 h drug treatment at five different concentrations, as shown in Fig. 3K and Supplementary Table 09. In agreement with DREEP predictions, HER2⁻ cells were much more sensitive to etoposide than HER2⁺ cells, while afatinib was equally effective on both subpopulations. This counterintuitive result was similar to that observed by Jordan et al[60] using circulating tumour cells from a BC patient sorted into HER2⁻ and HER2 + subpopulations, which were found to be equally sensitive to Lapatinib (another HER2 inhibitor), but no mechanism of action was put forward.

We hypothesize that the dynamic interconversion of MDA-MB-361 cells between the HER2⁻ and the HER2⁺ state may explain this surprising result: when the starting population consists of HER2⁻ cells only, some of these cells will nevertheless interconvert to HER2⁺ cells during afatinib treatment, and they will thus become sensitive to HER2 inhibition, explaining the observed results. We mathematically formalized this hypothesis with a simple mathematical model (Supplementary Figs. 21-23 and in the Supplementary Note 01) where two species (HER2⁺ and HER2⁻ cells) can replicate and interconvert, but only one (HER2⁺) is affected by afatinib treatment. The model shows that if the interconversion time between the two cell states is comparable to that of the cell cycle, then afatinib treatment will have the same effect on both subpopulations. If instead the interconversion time is much longer than the cell cycle, then afatinib will have little effect on HER2⁻ sorted cells, but maximal effects on HER2⁺ sorted cells, and vice-versa, if the interconversion time is much shorter than the cell cycle, then afatinib's effect would be minimal on both HER2⁻ and HER2⁺ sorted cells.

Comparison of the modelling results with the experimental results thus suggests that the interconversion rate should be of the same order of the cell cycle (about 72 h for MDAM361 cells). The model further predicts that treating the unsorted population of MDA-MB-361 cells with afatinib reduces the percentage of HER2⁺ cells, since only HER2⁺ will be affected, but that this percentage quickly recovers once Afatinib treatment is interrupted (Supplementary Figs. 22 and 23 and Supplementary Note 01).

To test modelling predictions, we treated the MDAM361 cell line (without sorting) with afatinib and etoposide and then assessed by cytofluorimetry the percentage of HER2+ and HER2⁻ cells before and after the treatment. As shown in Fig. 3L, M, and Supplementary Table 10 and Supplementary Table 11, etoposide increased the percentage of HER2⁺ cells, in agreement with the increased sensitivity of HER2⁻ cells to this treatment, whereas afatinib strongly decreased the percentage of HER2⁺ cells, confirming that its effect is specific for HER2⁺ cells only. We next measured the percentage of HER2⁺ cells following removal of Afatinib from the medium; as shown in Fig. 3N, O the percentage of HER2 + cells quickly increased confirming the modelling results. We next investigated the effect of Afatinib and Etoposide in combination in MDA-MB-361 cells. Specifically, we

tested 20 different combinations in triplicate experiments and measured cell viability in response to the treatments, as summarized in Supplementary Fig. 24A and Supplementary Data 05. We then used this dataset to estimate whether these two drugs had an additive, synergistic or antagonistic effect (Supplementary Fig. 24B). Overall, the average synergy score across all the combinations, measured using the Excess over Bliss model[67], is compatible with an additive effect (synergy score of −12.0 with a confidence interval of ±4.07 thus falling in the interval from −10 to +10 considered as additive[68]); however, for low concentrations of afatinib and high concentrations of etoposide, we did observe an unexpected tendency for the drugs to be antagonistic (indicated as yellow/red squares in Supplementary Fig. 22B). This inhibitory effect may be partly explained by the fact that anti-HER2 treatment in HER2⁺ cancer cells has been shown to downregulate the expression of TOP2A as well as of other genes involved in the G2-M cell cycle phase[69]. This may cause desensitization to Etoposide treatment, as it acts primarily on TOP2A during the S and G2 phases of the cell cycle[70].

Altogether our results show that DREEP can predict drug sensitivity from single-cell transcriptional profiles and that dynamic heterogeneity in gene expression does play a significant role in how the cell population will respond to the drug treatment.

## Discussion

In this study we provide a transcriptional characterization at single-cell level of a panel of 32 breast cell lines. We show that single-cell transcriptomics can be used to capture the expression of clinically relevant markers. Our approach could be very useful for automatically identifying gene signatures for less studied tumours for which no signature is currently available, and no clear clinical subtypes have been identified. We also show that breast cancer cell lines express clinically relevant BC receptors heterogeneously among cells within the same cell line. Moreover, we observed dynamic plasticity in the regulation of HER2 expression in the MDA-MB-361 cell line with striking consequences on drug response. This phenomenon has been recently observed also in circulating tumour cells of a BC patient[60] and in other cell lines[17,71].

We determined the cell line composition of patients' biopsies both from both single-cell and bulk gene expression profiles. Estimation of cancer cell-line composition provides an alternative and more information-rich framework to link bulk gene expression measurement of patient's biopsies to preclinical cancer models. Knowledge of drugs to which cancer cell lines are sensitive may also inform drug treatment for patients for which bulk gene expression profiles have been measured. However, further work is needed to assess the clinical relevance of these findings.

Single-cell transcriptomics is still not clinically ready because of the costs and time required. This work, however, shows the importance of performing single-cell sequencing on the available cancer models, including cell lines and organoids, to build a set of cell cancer states with a known phenotype and drug response to which patients' tumour can be mapped to make a leap in personalized diagnosis, prognosis and treatment of cancer patients.

## Methods

**Cell culture**. The 32 cell lines used in this study were obtained from commercial providers and cultured in ATCC recommended complete media at 37 °C and 5% CO₂. Cell-line identity was assessed through STR profiling by means of the AmpFlSTR Identifier Plus PCR Amplification kit (Applied Biosystems) with purified genomic DNA (1 ng) following the manufacturer protocol. KPL-1 cell line used in this study is indeed the same as the MCF7 cell line as previously reported (https://iclac.org/databases/cross-contaminations/).

**DROP-seq platform set-up**. Single-cell transcriptomic of the 32 cell lines was performed by implementing in-house the DROP-seq technology[20]. The microfluidics device for the generation of the droplet was fabricated using a bio-compatible, silicon-based polymer, polydimethylsiloxane (PDMS) that was rendered hydrophobic with Aquapel® treatment as per protocol[20]. In each sequencing experiment, cell suspension, bead suspension and carrier oil (QX200 droplet generation oil, Bio-Rad) were first loaded in syringes and then placed in syringe pumps (Leafluid). Flow rates of syringe pumps were set at 4,000 μL/h for both cell and barcoded bead suspensions while carrier oil syringe pump was set at 15,000 μL/h. In each sequencing experiment, cells and barcoded beads were, respectively, diluted at the concentration of 200 cell/μL in PBS with BSA 0.01% (Merck) and 120 bead/μL in lysis buffer. A self-built magnetic stirrer system was used to keep in suspension barcoded beads. To count the occurrence of a single cell together with a barcoded bead several tests were performed without lyses buffer in the bead suspension. In these tests, we observed about 5% of generated droplets filled with just one bead and one cell.

**Single-cell RNA library preparation and sequencing**. For each sequencing experiment, the targeted number of cells to sequence was set to 2000. Droplets were collected in a 50 mL falcon and broke by adding 1 mL of Perfluoro-1-octanol. Captured RNA was reverse transcribed in a single reaction following the original protocol[20] and then digested with exonuclease 1 to degrade unbound primers. Next, cDNA was first amplified with a total of 12 PCR cycles and then purified using AMPure XP beads at 0.6× ratio. Finally, the quality of the resulting cDNA library was quantified with the BioAnalyzer High Sensitivity DNA Chip and its concentration measured using the Qubit Fluorometer. The Illumina Nextera XT v2 kit was used to produce the next-generation sequencing (NGS) libraries using four aliquots of 600 pg of each cDNA library. Quality and concentration of NGS libraries were respectively quantified on the BioAnalyzer High Sensitivity DNA Chip and Qubit Fluorometer. Finally, either Illumina NextSeq 500/550 or NovaSeq 6000 machines were used to sequence the produced NGS libraries (Supplementary Table 01). Samples processed with NextSeq500/550 NGS library were diluted at the final concentration of 3 nM and sequenced using the 75-cycle high output flow cell while samples processed with NovaSeq 6000 machine were diluted at the final concentration of 250 pM and sequenced using the S1 100 cycles flow cell.

**Read alignment and gene expression quantification**. Raw data processing was performed using the Drop-seq tools package version 1.13 and following the Dropseq Core Computational Protocol (http://mccarrolllab.org/dropseq). Briefly, raw sequence data were filtered to remove all read pairs with at least one base in their cell barcode or UMI with a quality score less than 10. Then read 2 was trimmed at the 5' end to remove any TSO adapter sequence, and at the 3' end to remove polyA tails. Reads were then aligned using STAR[72] on hg38 human genome (primary assembly, version 28) downloaded from GENCODE[73]. After reads alignment, UMI tool[74] was used to perform UMI deduplication and quantify the number of gene transcripts in each cell. The initial number of sequenced cells was identified using a simple (knee-like) filtering rule as implemented by CellRanger 2.2.x. After this, only high depth cells with at least 2500 UMI, more than 1000 captured genes and with less than 50% of reads aligned on mitochondrial gene were retained. Putative multiples among the sequenced cells of each BC cell line were simply discarded identifying outliers in the count depth distribution by using Tukey's method based on lower and upper quartiles with k equal to 3. To check for the possibility of batch effects in the sequencing data, the counts of each gene in every single cell were summed overall the cells in the same cell line to obtain one pseudo-bulk sample per cell line, for a total of 32 pseudo-bulk samples. These samples were then normalized with EdgeR normalization[75] and a Principal Component Analysis (PCA) plot was performed and reported in Supplementary Fig. 1B. Visual inspection of the PCA plot confirmed the absence of major batch effects in the data.

**BC atlas construction**. Single-cells expression profiles were normalized using GF-ICF (Gene Frequency—Inverse Cell Frequency) normalization using the gficf package[76,77] for R statistical environment (https://github.com/dibbelab/gficf). GF-ICF is based on a data transformation model called the term frequency-inverse document frequency (TF-IDF) that has been extensively used in the field of text mining. GF-ICF transformation was applied on CPM (count per million) after EdgeR normalization[75] and discarding genes expressed in less than 5% of the total number of sequenced cells. Finally, each cell was summarized with its first 10 Principal Components (PCs) and projected with UMAP[78] into a two-dimensional embedded space. The number of principal components was chosen as the "elbow" point on the plot of the first 50 PCs. UMAP projection was performed by using the uwot package in the R statistical environment 3.6.

**Quantification of basal-like transcriptional profiles of triple-negative BC cell lines**. Genes known to be specifically expressed in basal epithelial cells were retrieved from the literature[21,79–84] and used to perform Gene Set Enrichment Analysis (GSEA) against the pseudo-bulk profiles of the 15 triple-negative BC cell lines in the atlas. Pseudo-bulk profiles for each cell line were obtained by summing the counts of each gene in every single cell overall the cells in the same cell line. The Enrichment Score from GSEA and its associated p-value are then used to assess the

extent to which each cell line expresses basal-like biomarkers. The results of this analysis are reported in Supplementary Table 03 and show that 11 out 15 triple-negative cell lines significantly ($p < 0.05$) express the basal biomarkers.

**Cell clustering and identification of marker genes**. Transcriptionally similar subpopulations of cells were found using a Phenograph like approach[85] as implemented in the clustcells function of gficf package[76]. Briefly, we initially built a graph of cells by using the K-Nearest Neighbours (KNN) algorithm applied to the PC-reduced space where each cell was connected to its 50 most similar cells using the manhattan distance. Then, to build the final graph of cells, the edge weight between any two cells was computed as the Jaccard similarity, i.e. the proportion of neighbours they share. The Louvain algorithm with a resolution parameter equal to 0.25 was used to find communities of cells in this graph. Differentially expressed genes in each cluster were identified by the findClusterMarkers function of gficf package, which compares the expression of a gene in each cluster versus all the other by using the Wilcoxon rank-sum test[76].

**TGCA bulk expression dataset and cell-line deconvolution**. Raw bulk expression data and relative patient clinical information were collected from the Genomic Data Commons (GDC) portal[86] by using the TCGAbiolinks package[87]. Then, raw counts were normalized by the EdgeR package[75] into R statistical environment 3.6. Bisque tool[55] (available at https://github.com/cozygene/bisque) was used to estimate the cell-line composition from the patient's bulk gene expression profile. Specifically, we applied the ReferenceBasedDecomposition function with parameters: bulk.eset set to the bulk gene expression dataset in log2 scale; sc.eset set to our single-cell BC atlas with normalized raw counts rescaled in log2; use.overlap set to FALSE and markers set to the marker genes across the 32 BC cell lines estimated by using the function findClusterMarkers of gficf package. As in the original manuscript describing the Bisque tool[55], only marker genes with an FDR < 0.5 and Log2 fold change greaten then 0.25 were used for deconvolution purpose.

**Spatial sequencing data**. Spatial transcriptomic data of two BC patients were download from 10× Genomic website (https://www.10xgenomics.com/resources/datasets). Only tiles reported to be "in tissue" according to the related metadata of each patient slide were used.

**Single-cell data of TNBC patients**. Pre-treatment single-cell data of the five TNBC patients[50] described in Fig. 02A, B were downloaded from GEO repository (accession number GSE148673). Then genes expressed in less than 5% of total cells across the five patients were filtered out. Finally, the raw UMI count matrix was normalized with edgeR package in R environment.

**Mapping new cells into the BC atlas and estimation of the cancer subtype**. New points were mapped to the UMAP space via embedNewCells function of gficf package[76] as depicted in Supplementary Fig. 07. Briefly, scRNA-seq profiles (or tiles from 10× spatial transcriptomics dataset) were normalized with gficf package using the ICF weight estimated on the BC atlas. Then scRNA-seq profiles were projected to the existing PC space using gene loadings from the BC atlas, via the umap_transform function of uwot package, which uses the UMAP estimated model to map the new cells into the existing UMAP space. Finally, the cancer subtype of each mapped cell was predicted with the function classify.cells of the package gficf with the k-nearest-neighbour parameter set to 100. The number of k-nearest-neighbours to use was chosen by computing the average classification of the method accuracy as a function of the number of neighbourhoods used using the cells of our breast cancer atlas. Specifically, 75% of cells in each cell line were collected and used as the training set (i.e. 26,455 cells) while the remaining 25% was used as test set (i.e. 8821 cells). Then, the 26,455 cells of the training set were used to reconstruct the breast cancer atlas from scratch. While the 8821 cells of the test set were mapped into the atlas as "new cells" with our mapping algorithm. Finally, the cell line type of each cell in the test set was predicted by using k-nearest-neighbours ranging from 1 to 300 (Supplementary Fig. 09B). Visual inspection of the plot shows the best performance of the method is obtained around k equal to 100.

**Estimation of heterogeneity in biomarker expression**. When determining whether a gene is truly heterogeneously expressed in single-cell RNA-seq data it is necessary to account for the probability of detection given the Poisson sampling of sequencing data[58]. To this end, for each cell line, we first calculated the expected proportion of zeros across cells for each of the four clinical biomarkers assuming a Poisson distribution of counts, by considering the heterogeneity in sequencing depth, according to this equation:

$$P^0_{x,i} = \text{Poisson}\left(0, \lambda_i\right) \tag{1}$$

where: $P^0_{x,i}$ is the probability for gene $x$ of not being detected in cell $i$, i.e. to have a zero UMI count; $\lambda_i$ is the expected number of counts for gene $x$ in cell $i$. To compute $\lambda_i$ in each cell we used the following equation:

$$\lambda_i = \langle UMI_x \rangle \cdot UMI^i / \langle UMI \rangle \tag{2}$$

where $\langle UMI_x \rangle$ the average UMI count of gene $x$ across the single cells, $UMI^i$ is the total UMI counts in cell $i$ and $\langle UMI \rangle$ is the average number of total UMI across cells. Using this model, we tested whether the measured zero proportion was higher than the expected rate under the Poisson model, as zero inflation indicates the presence of cell heterogeneity[59]. For each cell line, we computed an empirical p-value for each of the four biomarkers, by randomly sampling from N (number of cells in the cell line) Poisson distributions using the estimated $\lambda_i$. We thus obtained a "simulated" vector of counts, from which we computed the proportion of zero counts. This process was repeated 10,000 times to obtain an empirical distribution of the proportion of zero counts, which we then used to compute the empirical p-value. The results are reported in Supplementary Table 04.

**Correlation between drug targets expression and drug potency.** By using CTRPv2 and GDSC dataset we built a list of 302 drugs for which the target genes are known. Then, for each drug we correlated its reported potency with the percentage of cells expressing its target genes across our 32 cell lines (Supplementary Data 02). A gene was considered expressed if and only if at least one UMI was detected. In Fig. 3G only significant correlation values ($P < 0.05$) are plotted.

**Description and validation of the DREEP method for single-cell drug sensitivity prediction.** The naïve gene expression profiles (RNA-seq) of about 1000 cancer cell lines and the drug potency of each drug in each cell line, quantified by the Area Under the Curve (AUC) of the dose-response curve, are part of the CTRPv2 dataset publicly available from the Cancer Cell Line Encyclopaedia (CCLE) portal[24]. One hundred sixty-six cell lines belonging to liquid tumours were discarded and only 658 cell lines belonging to solid tumours were retained and used for further analysis. The raw counts of each gene were normalized with edgeR package[75] and transformed in log10(CPM + 1). Poorly expressed genes and genes whose entropy was in the fifth percentile were excluded from the analysis. Expression profiles of the 658 CCLs were then crossed with drug sensitivity data[8]. This dataset was originally composed of 481 small molecules, but, after removing drugs for which the in vitro response was available for less than 25 CCLs, only 450 small molecules were retained for further analysis. As schematized in Fig. 3H, for each gene and for each of the 450 drugs, we computed the Pearson correlation coefficient (PCC) between the expression of the gene across the 658 cell lines and the effect of the drug expressed in terms of AUC. Since the AUC reflects the in vitro response of a cell line to different concertation of a drug in a timeframe of 72 h, lower values of AUC are associated with sensitivity whereas higher values with resistance to the drug. Hence, genes positively correlated with the AUC are potential markers of resistance (the more expressed the gene, the higher the concentration needed to inhibit growth), vice-versa, negatively correlated genes are markers of sensitivity. We this approach, we generated a ranked list of expression-based biomarkers of drug sensitivity and resistance for each of the 450 drugs where genes positively correlated with the AUC are at the top, and those negatively correlated at the bottom. Finally, to predict drug sensitivity at the single-cell level, we used the top 250 expressed genes of each cell as input of Gene Set Enrichment Analysis (GSEA)[63] against the ranked list of biomarkers for each one of 450 drugs built as described above (Fig. 3I). Hence, while a negative enrichment score implies that genes associated with drug sensitivity are highly expressed by the cell, a positive one indicates the cell express genes conferring drug resistance. GSEA and associated p-values were estimating using the fgsea package in the R statistical environment version 3.6. To assess the precision and sensitivity of DREEP in predicting drug response from single-cell transcriptional profiles, we evaluated its performance on two publicly available drug screening dataset: one derived from the CTRPv2 dataset[24] and the other derived from Genomics of Drug Sensitivity in Cancer (GDSC) study by the Sanger Institute[9], which includes drug potency data measured as IC50 for about 250 small molecules (of which only 86 in common with the CTRPv2 dataset). To build the "CTRPv2 golden standard" for 450 drugs, we first computed the z-score percentiles from the AUC of each drug across all the 824 cancer cell lines. We then defined a cell line sensitive to the drug if and only if its Z-score was in the 5% percentile. The "CTRPv2 golden standard" for the 32 cell lines in the atlas was built by assigning to each of $32 \times 450$ (=14,400) cell line/drug pair the value 1 if the cell line was sensitive to the drug and 0 otherwise. To build the "GDSC golden standard" of $32 \times 86$ drugs (=2752), we set a specific threshold for IC50 to call a cell line sensitive to a drug as previously described[7] and assigned to each cell line/drug pair the value 1 if the cell line was sensitive to the drug and 0 otherwise. We then applied DREEP to the single-cell profiles of the 32 BC cell lines to predict the percentage of sensitive cells in each cell line for the 450 drugs. Finally, Positive Predicted Values (PPV) were defined as TP/(TP + FP) where TP represents the number of true positives and FP the number of false positives predicted cell lines/drug pairs.

**Estimation of classification accuracy of PAM50, scCCL or scCCL + HER2 signatures on TCGA patients.** We divided the set of 937 patients from TGCA, for whom cancer subtype was annotated, into a training set of 625 patients (two-thirds of the patients) and a test set of 312 patients (one third of the patients). The training set was used to train the classifier algorithm (XGBoost) with the chosen gene signature (PAM50, scCCL or scCCL+HER2) while the test set was used to compute the classification accuracy (the percentage of patients correctly classified) for each tumour subtype. We repeated this process three times (i.e., 3-fold cross-validation), each time randomly assigning patients to the training set and to the test set and then computing the classification accuracy.
PAM50 signature was downloaded from the original publication and converted in ensemble id before being used. While XGBoost model was trained by using xgboost function of xgboost R library.

**Cell2location analysis.** Cell to location tool was run with default parameters and following the tutorial at https://cell2location.readthedocs.io/en/latest.

**Drug sensitivity of the HER2+ and HER2− subpopulations in the MDA-MB-361 cell line.** For each sequenced cell of the MDA-MB-361 cell line, the enrichment score of 450 anticancer drugs was predicted as described above. Then, to identify drugs exhibiting differential sensitivity for the two subpopulations, we used the Mann-Whitney test was to assess if there was a difference between the enrichment scores of HER2+ and HER2− subpopulations. P-values were corrected for false discovery rate using Benjamini-Hochberg correction. A drug was considered specific for HER2− cell population if and only if its FDR was less than 0.05 and the median enrichment score across HER2− cells less than zero while its median enrichment score across HER2 + cells greater than zero. Conversely, a drug was considered specific for the HER2+ cell population if and only if FDR was less than 0.05 and its median enrichment score across HER2+ cells less than zero while its median enrichment score across HER2− cells greater than zero.

**Prediction of cell cycle phase from scRNA-seq.** The cell cycle phase of each sequenced cell was predicted using the function *CellCycleScoring* of the *Seurat* tool with default parameter and following what was suggested in the corresponding vignette (https://satijalab.org/seurat).

**HER2 antibody staining procedure for flow cytometry analysis.** Cells were first washed with phosphate-buffered saline (PBS) 1×, detached with 0.05% trypsin-EDTA, resuspended and harvested with the appropriate medium in single-cell suspension. Then, cells were counted, washed with PBS-FBS 1%, and finally incubated for 15 min at 4° in the dark at the concentration of $1.0 \times 10^6$ cell/µL with staining buffer. The staining buffer was prepared to dilute the mouse anti-human HER2 antibody (BD BB700) at the final concentration of 0.00114 ng/µL. Then, to remove the unbound antibodies, cells were washed three times with PBS-FBS 1%. Flow cytometry measurements were performed on either BD Accuri C6 or BD FACSAria III instruments. To define antibody positive and negative cells, the unstained samples were used to set the gate. To record data, at least $1.0 \times 10^4$ events were collected for each sample. Data analysis was performed using either BD FACSDiva 8.0.1 or BD Accuri C6 software.

**HER2 expression dynamics experiment.** Sorting of MDA-MB-361 HER2-positive and HER2-negative cells was performed following the antibody staining procedure described above with the only exception that before sorting, each sample was resuspended in sorting buffer (PBS 1×, FBS 1%, trypsin 0.1%, EDTA 2 mM). Then, $4.0 \times 10^5$ cells were collected for each cell subpopulation (i.e. HER2-positive and HER2-negative), plated in their appropriate medium, and incubated at 37 °C. After 18 days, the percentage of cells expressing HER2 protein was checked by performing the antibody staining procedure described above.

**Drug sensitivity assay.** Cells were seeded in 96-well microplates (PerkinElmer); the seeding cell confluency was specifically optimized for each cancer cell line to have cells in a growth phase at the end of the assay. After overnight incubation at 37 °C, cells were treated with DMSO (Merck) for the negative control and with five concentrations of selected drugs in triplicate. Cells were then incubated at 37 °C for 72 h. Cell viability was assessed by measuring either luminescence with GloMax® Discover instrument from Promega or by nuclei count using the Operetta instrument from PerkinElmer. Luminescence measurements were normalized using background wells as manufacturer protocol. For luminescence measurement, cells were treated with Promega CellTiter-Glo® Luminescent Cell Viability Assay according to the manufacturer protocol. For nuclei count, cells were washed with PBS 1×, fixed with paraformaldehyde (PFA) 4% for 10 min at room temperature, washed with PBS 1×, incubated at room temperature in the dark with HOECHST 33342 (Thermo Fisher Scientific) diluted 1:1000 in PBS 1× for 10 min and finally washed with PBS 1×. Nuclei count was performed using Columbus image analysis software (PerkinElmer). All drugs used in this study were purchased from Selleckchem.

**Drug combination assay.** To perform the drug combination assay, afatinib and etoposide were first prepared in five dilutions as a single agent. Then, from the single-agent dilutions, afatinib and etoposide were combined in all possible dose combinations, generating a $4 \times 5$ (i.e. afatinib × etoposide) drug pair matrix. MDA-MB-361 cells were seeded in 96-well plate and incubated as described above. Then, cells were treated in triplicate with single-agent afatinib and etoposide and with the drug pairs of the $4 \times 5$ matrix. In addition, DMSO was used in triplicate as a negative control of the drug treatment. Following 72 h of drug incubation, cell

viability was measured with the Promega CellTiter-Glo®Luminescent Cell Viability Assay, as described above. A total of three independent drug combination assays were performed, and in each assay the luminescence data of replicates were averaged. The expected drug combination responses were calculated using Syner-gyFinder version 2.0[68], based on the Bliss model. The input file for SynergyFinder was generated including the viability data of each independent assay.

## Data availability

The single-cell BC data generated in this study have been deposited in the Gene Expression Omnibus (GEO) database under accession code GSE173634. Raw counts matrix stored as R object or matrix market format of the 35,276 cells from which the BC atlas was built are also available on figshare at following https://doi.org/10.6084/m9.figshare.15022698 [https://figshare.com/articles/dataset/Single_Cell_Breast_Cancer_cell-line_Atlas/15022698]. Bulk cancer cell line gene expression, mutation and copy number alteration dataset used in this study are publicly available through depmap portal at [https://depmap.org/portal]. Breast spatial transcriptomic data are available from 10× data portal at [https://www.10xgenomics.com/resources/datasets]. Cell-line drug screening datasets used in this study are publicly available from cancerrxgene portal at [https://www.cancerrxgene.org/] and the cancer therapeutics response portal at [https://portals.broadinstitute.org/ctrp.v2.1/]. All other relevant data supporting the key findings of this study are available within the article and its Supplementary Information files. Source data are provided with this paper.

## Code availability

The code[88] to reproduce the BC atlas from raw counts is available on GitHub dibbelab/singlecell_bcatlas [https://github.com/dibbelab/singlecell_bcatlas]. Moreover, the single-cell atlas can be explored at http://bcatlas.tigem.it.

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

## Acknowledgements
This work was supported by the AIRC (Associazione Italiana Ricerca sul Cancro) Grant IG 2016-18479 and IG 2021-26161 and by iPC project H2020 826121. G.G. was supported in part by the STAR (Sostegno Territoriale alle Attività di Ricerca) grant of University of Naples Federico II.

## Author contributions
G.G. performed all computational analysis, conceived the method for single-cell drug sensitivity prediction and contributed to the writing of the manuscript. G.V. performed all the experiments including setting up the DropSeq platform, single-cell RNA-sequencing and drug response validations. B.T. performed cytometric analyses and supported G.V. in cell culture and RNA-seq library preparation. A.I. and R.B. contributed to data discussion and writing of the manuscript. D.d.B. supervised the work, contributed to the writing of the manuscript and conceived the original idea.

## Competing interests
The authors declare no competing interests.
