## [Peer review file · Nature Communications]

REVIEWER COMMENTS

Reviewer #1 (Remarks to the Author): Expert in drug response prediction and bioinformatics

Gambardella et al. describe the construction of a single cell breast cancer cell line atlas in this manuscript. The authors sequenced >30,000 single cells from different breast cancer cell lines, and used this dataset to categorize cancer cells from patient scRNA-seq data and also spatial transcriptomics datasets. They compared the single cell expression profiles with drug sensitivity, and developed the DREEP method to predict drug sensitivity from scRNA-seq data. Interestingly, they found substantial heterogeneity even in the "homogeneous" cell lines, and connected this heterogeneity to drug sensitivity. They also present their results as an interactive web application, which fosters further analysis.

Investigating intra-tumor (or in this case intra-cell line) heterogeneity and connecting it to drug sensitivity is a very important aspect of current oncology bioinformatics, so the manuscript can be interesting for the field. The authors collected a nice dataset and used / developed methods to connect the cell line single cell dataset to patient level (single cell, spatial and bulk) transcriptomic data. The single cell RNAseq analysis and methods to compare with patient data sound solid and the results are convincing. I have several questions regarding the drug sensitivity analysis (which is more my expertise).

1) The authors compared (Figure 3F) the percent of EGFR/ERBB2 expressing cells (single cell atlas) to the EGFR/ERBB2 sensitivity of these cell lines (from public, "bulk" sensitivity datasets) and found an expected correlation between drug sensitivity and target expression. How does this correlation compare to the correlation of bulk EGFR/ERBB2 expression (from CCLE or other bulk cell line expression dataset) and drug sensitivity? If the correlation on single cell level is higher, that would underline the importance of single cell level analysis. Also, how compares the general correlation between percent of target expressing cells / drug sensitivity (Figure 3G) to bulk expression / drug sensitivity?

2) The authors developed the DREEP method, which is an enrichment based method to predict drug sensitivity from scRNAseq data. They evaluated the performance of DREEP on 86 drugs in Figure 3J using Positive Predictive Value. How did they select these 86 drugs? Could the authors show some more widely used evaluation metric (ROC curve or precision-recall curve (PRC), with corresponding "random model") for testing the predictive performance of DREEP? Also, it would be interesting to see (at least on the level of ROC / PRC AUCs) the drug-wise performance of DREEP, not just for the whole dataset.

3) The identification of different drug sensitivity in subpopulations of MDA-MB-361 are very interesting results. As drug synergy is possible not only on the individual cell level, but also on the level of cell populations, these results suggest interesting applications for drug combinations. Is it possible that drugs selectively inhibiting different populations of the same cell line can act synergistically? Could the authors show some experimental evidence for this? (e.g.: Afatinib and Etoposide are synergistic in MDA-MB-361 / and in the different sub-populations?)

Bence Szalai

Reviewer #2 (Remarks to the Author): Expert in single-cell RNA-seq and bioinformatics

The authors present an atlas of single-cell data from 32 distinct breast cancer cell lines. They show the single-cell data recapitulates general differences between the lines that

have been previously observed. They identify 22 markers from their clusters and show they have some heterogeneity in expression in cancer patients from TCGA. They identify heterogeneity in HER2/ERBB2 expression between cells from the same line and show this has functional consequences with respect to drug responses. The paper is written clearly, and all data and computational tools are publicly available.

Major Comments (divided by results section for clarity):

(1) In non-linear dimensionality reduction the distances between distinct clusters are typically distorted thus describing a collection of distinct clusters as an "archipelago" implying they are likely similar to each other in some way is misleading. Cell-lines should be described as either distinct clusters or part of the luminal-supergroup.

Why didn't the authors identify CNVs or other genetic variants across their cell-line that could explain why certain lines form distinct clusters - i.e. because they harbour unique variants?

(2) Biomarkers:

How were the biomarkers for each cluster identified - what was the criterion used to choose these particular genes from the results of the Wilcoxon rank-sum tests? Why only 22 chosen? Could adding more biomarkers produce more accurate classifications and surpass the accuracy of the PAM50 markers?

Why couldn't MAGEA4 and XAGE2 be mapped to TCGA? Did the authors look for alternative names for these genes in the TCGA data? (MAGEA4 has 7 alternative symbols, XAGE2 has 4 alternative symbols)

(3) Mapping & Deconvolution:

When using a published tool such as Bisque, the tool name should be stated in the main-text along with the reference. Saying "we next trained a deconvolution algorithm" (line 175) is misleading and claiming credit for someone else's work - i.e. the creators of the Bisque tool.

The mapping tool is a new addition to the author's gfcf package that was not present in their original publication, thus the complete details of the algorithm should be included in the manuscript methods section. How does the novel mapping tool compare to existing published methods (e.g. scmap)?

When using their single-cell data to label spatial data why didn't they use existing deconvolution tools (e.g. cell2location)? Spatial data is not single-cell resolution thus each spot could be a mixture of profiles similar to multiple clusters in their data, deconvolution would be able to correctly quantify this mixture rather than attempting to assign the entire spot to one of the clusters.

Why did the authors choose to deconvolve bulk and spatial data by the cell-lines rather than by their clusters with conserved expression profiles across cell-lines?

For Figure 1H what was the statistical significance of the differential expression across TCGA cancer types?

For Figure 1I the classification task, is the accuracy coming from true positives or true negative classifications? Precision or Positive Predictive value would be a more informative

statistic in this case.

4) Heterogeneity:

When determining whether a gene is truly heterogeneously expressed in single-cell RNAseq data it is necessary to account for the probability of detection given the Poisson sampling of sequencing data. While Supplementary Figure 8 attempt to answer this it does not take into account the heterogeneity in sequencing depth of each individual cell. The authors should calculate probability of detection in each cell of each line and then average across those for each cell-line to determine its expected detection rate:

$$\text{prob_detect_cell_i} = \text{Poisson}(0, \text{lambda} = \text{mean_umi_per_cell_gene_X} * \text{total_umi_cell_i} / \text{mean_total_umi_per_cell})$$

When the authors consider the cell-cycle they use a tool that classifies all cells into G1 or S/G2M, this excludes the possibility of cells having a senescent or non-cycling phenotype. Could the authors determine what portion of their cells are not cycling at all?

Figure 3E Were these pathways the most significantly enriched? If not out of how many total pathways with FDR < 10% were they selected from? How were they selected? Figure 3D shows very little difference in proportion of G2M cells, thus it seems contradictory for cell-cycle to be the most enriched pathway differentially expressed between the two populations. Could the authors comment on why this is the case?

(5) Drug response

In the main text it is stated the drug sensitivity was estimated for HER2 inhibitors, however the methods section explaining this analysis makes no mention of filtering for different drug mechanisms. Was the analysis considering all drugs in the database or only HER2 inhibitors? For Figure 3F is this showing direct drug potency from the CCLE database or inferred drug potency from the DREEP model?

In addition the explanation of the novel DREEP method is confusing, as this is a novel method please provide a fully detailed and clear description of how this method works. Is the section titled: "Single-cell drug sensitivity prediction" describing the DREEP method? For the validation of this tool why did the authors use a Z-score percentile - this would force cell-lines to only be recorded as sensitive to a small number of drugs even if in truth they are highly sensitive to all the tested drugs. IC50 values are standardized across lines, it would be more appropriate to choose a specific IC50 value threshold to determine if cell-lines are or are not sensitive to the drug for their gold-standard.

Reviewer #3 (Remarks to the Author): Expert in breast cancer genomics

I read the manuscript by Gambardella and colleagues with interest.

In this work, the authors profiled the transcriptome of 35,276 cells from 32 breast cancer cell lines to create a breast cancer cell line atlas.

A high degree of heterogeneity was found across individual cells within the same cell line. Furthermore, single cell transcriptomic profiles from patient tumor biopsy samples were compared to the atlas and found to be heterogenous.

Lastly, results from in vitro drug screening were used to predict responses to anticancer agents, providing a new resource for drug response determination.

Major comments:

- 1) The interpretation of the results of this paper heavily relies on the 32 cell lines selected. More information about each cell-line should be detailed. How did the investigators ensure that the cell-lines have not been cross contaminated, and that the cluster results truly reflects tumor heterogeneity? STR profiling or other genomic analyses could be useful for this purpose.**
- 2) For the single cell atlas generation, the authors should justify their approach of not doing batch effect correction, especially since the experiment involves two different sequencing platforms.**
- 3) Line 87, for Fig 1D, the authors should also include the relative expression of these genes, instead of just showing the percentage of cells expressing them, to better clarify the phrase "highly expressed".**
- 4) Line 88-89, the authors should quantify the similarity between triple-negative BC cell lines with basal-like cell to support their statement, in addition to listing the common genes expressed.**
- 5) Lines 125-127, it is hard to distinguish LumA, LumB, and maybe Her2 from just the 20 biomarkers being derived. The findings shown in Supplementary Figure 4 are not adequate to support the statement made.**
- 6) Lines 127 - The authors cited that the expression of KLK10 in a TNBC subset represents an opportunity for biomarker discovery. However, neither tamoxifen nor trastuzumab are used in TNBC, and thus the relevance of KLK10 in TNBC is questionable.**
- 7) Lines 131 - The presence of ACTG2 in a subset of TNBC and its relationship to cell proliferation and platinum-based chemosensitivity is highlighted here. The authors should provide evidence for this statement here using the cell-lines available e.g. HS578T/MX1.**
- 8) Lines 133 - The performance of the 20 genes against the PAM50 gene signature was deemed to be "overall comparable". This statement should be backed up by appropriate statistical tests.**
- 9) The authors derived a methodology to assign cell line data to patient tumor profiles, starting with single cell profiles of 5 TNBC cases. Curiously 1 case - TNBC5 was mapped as "luminal" instead. No further information on the 5 TNBC samples were provided, including how the original diagnosis was made. Similarly, spatial transcriptomics of 2 cases were used to test the algorithm, picking up HER2-overexpressing or triple negative cell populations. Corresponding IHC and HER2 FISH data should be demonstrated to demonstrate accuracy of the original diagnoses.**
- 10) The assignment of specific tumor cell lines to the composition of whole tumor samples (using bulk transcriptomic data), while interesting, needs further validation with corresponding single cell data from the same tumor samples.**
- 11) Fig 1H, the authors should do a statistical test between PAM50 and scCCL markers. If scCCL markers are not doing better when compared to PAM50, then why should they be**

used?

**12) Line 183 - "JIMT-1 is an HER2-overexpressing cell line with an amplified 184 ERBB2 locus, but no HER2+ patient was mapped to this cell line"
Figure 2E however seems to show that JIMT-1 maps to HER2+ patient TCGA-E2-A1B0**

13) For the cancer subtype classification section, the authors should show how frequent their algorithm will be able to predict tumor subtype correctly (may be using simulated data), given that the data from scRNAseq, spatial sequencing, and bulk RNAseq have showed some inaccurate classification to certain degree.

14) Line 202-204, the claim the authors made based on Supplementary Figure 8 require additional justifications / data, given that the p-value is so high.

15) At the last paragraph of Discussion, the authors had heavily emphasized how important the single-cell RNAseq is, but throughout the manuscript, there wasn't much details/analysis on getting the differential expression within the same tumor, instead most of the tumor subtypes were clustered according to their own origin (triple negative), or they were grouped together (luminal).

16) Overall, while the concept of using cell-line information as a base to infer heterogeneity in tumor samples is interesting, further validation is required. Ideally, a good validation set of cases should have both bulk transcriptomic and single cell gene expression data. Spatial transcriptomic data will be useful as well. Most crucially, the cases should be properly annotated histopathologically.

Minor comments:

1) In the Results section, the authors described luminal BC cells and TNBC cells forming "islands", "peninsulas" and "archipelagos". This is hard to visualize and it is unclear what the authors are trying to elucidate from this description. Perhaps figure annotations can be used to highlight this.

2) Figure 1F/Supplementary Figure 2 - colors used should be defined

3) Color definitions of LA, LB, H, TNA, TNB are sometimes different across figures. This makes interpretation of figures difficult.

4) Line 163, it should be Supplementary Figure 6B,C (instead of Supplementary Figure 6C,D)

5) Line 165, it should be Supplementary Figure 6B (instead of Supplementary Figure 6C)

6) Line 181, "With" instead of "We" this approach

Point-by-point response to Reviewers:

In what follows we replied to editor's and reviewers' comments and suggestions. We renumbered the comments to make cross-reference easier and we formatted in *italic* the reviewers' comments, whereas our replies are in normal text. We also highlighted changes made in the revised manuscript to make it easier for the reviewers to detect them.

Reviewer #1 (Remarks to the Author): Expert in drug response prediction and bioinformatics

Gambardella et al. describe the construction of a single cell breast cancer cell line atlas in this manuscript. The authors sequenced >30,000 single cells from different breast cancer cell lines, and used this dataset to categorize cancer cells from patient scRNA-seq data and also spatial transcriptomics datasets. They compared the single cell expression profiles with drug sensitivity, and developed the DREEP method to predict drug sensitivity from scRNA-seq data. Interestingly, they found substantial heterogeneity even in the "homogeneous" cell lines, and connected this heterogeneity to drug sensitivity. They also present their results as an interactive web application, which fosters further analysis. Investigating intra-tumor (or in this case intra-cell line) heterogeneity and connecting it to drug sensitivity is a very important aspect of current oncology bioinformatics, so the manuscript can be interesting for the field. The authors collected a nice dataset and used / developed methods to connect the cell line single cell dataset to patient level (single cell, spatial and bulk) transcriptomic data. The single cell RNAseq analysis and methods to compare with patient data sound solid and the results are convincing. I have several questions regarding the drug sensitivity analysis (which is more my expertise).

We really thank the reviewer for his/her comments and for appreciating our work.

1.1 *The authors compared (Figure 3F) the percent of EGFR/ERBB2 expressing cells (single cell atlas) to the EGFR/ERBB2 sensitivity of these cell lines (from public, "bulk" sensitivity datasets) and found an expected correlation between drug sensitivity and target expression. How does this correlation compare to the correlation of bulk EGFR/ERBB2 expression (from CCLE or other bulk cell line expression dataset) and drug sensitivity? If the correlation on single cell level is higher, that would underline the importance of single cell level analysis. Also, how compares the general correlation between percent of target expressing cells / drug sensitivity (Figure 3G) to bulk expression / drug sensitivity?*

Following the reviewer's suggestions, we have now computed the correlation between the expression of EGFR/ERBB2 in a cell line from its bulk expression (RNA-seq) from CCLE and the drug potency of anti-EGFR/ERBB2 drugs and reported these results in the new Suppl. Figure 16A. The correlation values obtained with bulk expression are comparable to those obtained with the percentage of EGFR/ERBB2 expressing cells. This result was somewhat expected since the percentage of cells expressing a gene from single cell RNA-seq is correlated with its average expression as measured in bulk RNA-seq (new Suppl. Figure 16B). Finally, we computed the correlation between percentage of cells expressing the cognate drug target and the potency of the drug across cell lines for additional 66 drugs using two different drug potency databases and compared it with those obtained from bulk gene expression, as shown in the new Suppl. Figure 16C. The results show that both bulk and single cell gene

expression produce very similar results. We have now modified lines 326-329 in the revised manuscript to better highlight this point.

1.2 *The authors developed the DREEP method, which is an enrichment based method to predict drug sensitivity from scRNAseq data. They evaluated the performance of DREEP on 86 drugs in Figure 3J using Positive Predictive Value. How did they select these 86 drugs? Could the authors show some more widely used evaluation metric (ROC curve or precision-recall curve (PRC), with corresponding “random model”) for testing the predictive performance of DREEP? Also, it would be interesting to see (at least on the level of ROC / PRC AUCs) the drug-wise performance of DREEP, not just for the whole dataset.*

We thank the reviewer for his/her suggestions. DREEP was trained on the CTRPv2 drug sensitivity dataset available in the Cancer Cell Line Encyclopedia (CCLE) from the BRAOD institute (1). In this dataset, the drug potency of 450 drugs across 658 cell lines is quantified by the Area Under the Curve (AUC) of the dose response measurements. However, in the original manuscript, we decided to report the results only for those 86 drugs (out of 450) that were also present in the Genomics of Drug Sensitivity in Cancer (GDSC) dataset from the Sanger Institute (2), which includes drugs sensitivity data for about 250 small molecules (of which only 86 in common with the CTRPv2 dataset). We had originally made this choice to have a more robust “golden standard”, i.e. validated by two independent studies against which to measure DREEP performance, however this “strict” choice is not necessary, and it is indeed possible to assess the performance of the algorithm for all 450 drugs. Hence, to comply with the reviewer’s suggestion, we have now validated DREEP’s performance against two “golden standards”, one obtained using the AUC data for 450 drugs from CTRPv2 dataset, and the other using the Inhibitory Concentration 50 (IC50) data for 86 drugs available in the GDSC dataset. The results are now shown in the revised Figure 3J for the CTRPv2 golden standard reporting and in the new Supplementary Figure 18 for the GDSC golden standard. Finally, we evaluated the drug-wise performance of DREEP method for the 450 drugs a new Supplementary Table 11. We have now revised manuscript’s main text to include description of these new results (lines 361-371) and updated the Methods’ section accordingly (Methods - lines 701-715).

1.3 *The identification of different drug sensitivity in subpopulations of MDA-MB-361 are very interesting results. As drug synergy is possible not only on the individual cell level, but also on the level of cell populations, these results suggest interesting applications for drug combinations. Is it possible that drugs selectively inhibiting different populations of the same cell line can act synergistically? Could the authors show some experimental evidence for this? (e.g.: Afatinib and Etoposide are synergistic in MDA-MB-361 / and in the different sub-populations?)*

We thank the reviewer for his encouraging comments. Indeed, the concept of synergy as acting not on the same cell but at the population level was the motivation for our single-cell drug prediction algorithm. For example, in the case of MDA-MB-361 cells, were the HER+ and HER- subpopulations differentially sensitive to both afatinib and etoposide, then a synergistic effect should be expected. However, since the population shifts dynamically between HER2- and HER2+ states, drug combination’s effects could be more complex. For example, we observed that Afatinib treatment for 72h acts equally on the two subpopulations (as shown in Figure 3K) and we had explained this effect in the original manuscript as caused by dynamic switching between HER2- and HER2+ states, so that even HER2- cells will become sensitive to Afatinib since they will become HER2+ during the treatment. Mathematical modelling of this behaviour supported our conclusions (Suppl. Fig 13 in the original manuscript). We have now performed experiments to probe the effect of Afatinib and Etoposide in

combination in MDA-MB-361, whose results are summarised in a new Supplementary Figure 24. Specifically, we tested 20 different combinations in triplicate experiments and measured cell viability in response to the treatments. Overall, we obtained an average synergy score, measured using the Excess over Bliss model, not significantly departing from the additive model (synergy score of -12.0 with a confidence interval of +/- 4.07 thus falling in the interval from -10 to +10 considered as additive); however, for low concentrations of afatinib and high concentrations of etoposide, we observed an unexpected tendency for the drug combination to be antagonistic (indicated as yellow/red squares in Supplementary Figure 24). This inhibitory effect may be partly explained by the fact that anti-HER2 treatment in HER2+ cancer cells has been shown to downregulate the expression of *TOP2A* as well as of other genes involved in G2-M cell cycle phase(3). This may cause desensitisation to Etoposide treatment, as it acts primarily on *TOP2A* during the S and G2 phases of the cell cycle(4). . Despite this inhibitory effect is indeed interesting, synergistic drug effects are typically difficult to investigate and we believe that further investigations are out of the scope of this publication. We have now added these new experimental data in a new Supplementary Figure 24 and a discussion of these findings in the revised manuscript (lines 419-434)

Reviewer #2 (Remarks to the Author): Expert in single-cell RNA-seq and bioinformatics

The authors present an atlas of single-cell data from 32 distinct breast cancer cell lines. They show the single-cell data recapitulates general differences between the lines that have been previously observed. They identify 22 markers from their clusters and show they have some heterogeneity in expression in cancer patients from TCGA. They identify heterogeneity in HER2/ERBB2 expression between cells from the same line and show this has functional consequences with respect to drug responses. The paper is written clearly, and all data and computational tools are publicly available.

We thank the reviewer for appreciating our work.

Major Comments (divided by results section for clarity):

2.1 *In non-linear dimensionality reduction the distances between distinct clusters are typically distorted thus describing a collection of distinct clusters as an "archipelago" implying they are likely similar to each other in some way is misleading. Cell-lines should be described as either distinct clusters or part of the luminal-supergroup. Why didn't the authors identify CNVs or other genetic variants across their cell-line that could explain why certain lines form distinct clusters - i.e. because they harbour unique variants?*

We agree with the reviewer that, in the pursuit of clarity, we ended up with a description of the clustering that was not rigorous. Indeed, by “archipelago” we meant that clusters were distinct from each other; we have now rewritten this section to avoid confusing the reader and adopted the wording suggested by the reviewer (revised manuscript lines 81-85). Moreover, as suggested by the reviewer, we clustered cell lines according to genomic variants (new Suppl. Figure 02A) and CNVs genomic obtained from the Cancer Cell Line Encyclopaedia (new Suppl. Figure 02B). We then mapped the CNV-based clusters onto to atlas, as shown in new Suppl. Figure 02C, to check whether cell-lines in the luminal-supergroup tend to fall in the same CNV-based clusters, and thus share similar CNVs, and vice-versa, whether cell lines in distinct single-cell clusters tend to be in distinct CNV-based cluster and thus have unique CNVs. As shown in new Suppl. Figure 02C, we found no obvious pattern, for example the CNV-based cluster 5 (cyan) contains three cell-lines (AU565, BT474 and T47D) with similar CNVs, however the Her2+

AU565 cell line forms a distinct cluster in single-cell atlas, while the luminal BT474 and T47D cell lines belong to the luminal-supergroup; similarly the CNV-based cluster 4 (blue) has three cell lines: CAL51, BT549 and HS578T, which form distinct single cell clusters, as shown in new Suppl. Figure 02B. As shown in Suppl. Figure 02A, when we tried clustering cell lines according to mutations, we found that, apart from a very strong similarity in mutational profiles between KPL1 and MCF7 cell lines, all the other cell lines are quite distinct from each other and thus no clustering is possible. These observations are now reported in the revised manuscript (lines 85-98).

2.2 Biomarkers: *How were the biomarkers for each cluster identified - what was the criterion used to choose these particular genes from the results of the Wilcoxon rank-sum tests? Why only 22 chosen? Could adding more biomarkers produce more accurate classifications and surpass the accuracy of the PAM50 markers?*

We apologise for not being clearer in the original manuscript. The 22 marker genes were chosen simply by selecting one gene for each of the 22 clusters. The gene selected in each cluster was the one most differentially expressed between cells in that cluster and all the remaining cells (lines 128-137 of revised manuscript). Nevertheless, as suggested by the reviewer, we also tried selecting 2 biomarkers per cluster thus obtaining a total of 44 biomarkers, hence comparable in number to the 50 biomarkers included in the PAM50, but we did not observe any significant gain in classification accuracy (data now shown). It is important to remark that the PAM50 signature was specifically designed to classify BC subtypes, whereas the signature we derived has no knowledge of breast cancer subtypes but was automatically generated by clustering single cell data of cell lines without any additional information. Our aim was not building a novel gene signature to classify BC patients, but rather to show that by clustering single-cell data it is possible to detect clinically relevant biomarker genes. Please also refer to our replies to point 3.5, 3.8 and 3.11 for additional analyses.

2.3 *Why couldn't MAGEA4 and XAGE2 be mapped to TCGA? Did the authors look for alternative names for these genes in the TCGA data? (MAGEA4 has 7 alternative symbols, XAGE2 has 4 alternative symbols)*

We thank the reviewer for this observation. The reason why MAGEA4 and XAGE2 were not mapped to TCGA dataset was caused by the way we pre-processed TGCA dataset. Specifically, during the pre-processing step, we had removed poorly expressed genes, i.e. those whose average expression across the 937 breast cancer patients from the TGCA collection was in the lower 5% percentile, and MAGE4 and XAGE2 were among these ones. However, we noticed that in a few patients these genes were indeed highly expressed. Therefore, we have now decided to remove this pre-processing step and included all the genes measured in TGCA in the analysis, thus recovering both MAGE4 and XAGE2. We have now modified Figure 1H and Figure 1I in the revised manuscript showing all the 22 biomarker genes from our BC atlas. Interestingly, these two biomarkers are highly expressed only in a subset of triple-negative breast cancer patients and of HER2+ /ER- patients; moreover, one of the two biomarkers (MAGE4) has been previously reported in the literature as overexpressed in HER2+ /ER- patients using proteomic profiling (5). We have now included these observations in the revised manuscript (lines 144-148)

2.4 Mapping & Deconvolution: *When using a published tool such as Bisque, the tool name should be stated in the main-text along with the reference. Saying "we next trained a deconvolution algorithm" (line 175) is misleading and claiming credit for someone else's work - i.e. the creators of the Bisque tool.*

We are sorry for this misunderstanding, obviously we did not want to take credit for this, but we understand that citing the name in the Methods only may appear unfair. As suggested by the reviewer, we have now revised the manuscript to include the name of the Bisque tool in the main text (lines 231-242 of the revised manuscript).

2.5 *The mapping tool is a new addition to the author's gfcf package that was not present in their original publication, thus the complete details of the algorithm should be included in the manuscript methods section. How does the novel mapping tool compare to existing published methods (e.g. scmap)?*

We thank the reviewer for his/her comments. We have now revised the manuscript (lines 174-182) and added a detailed explanation in the Methods section (paragraph “Mapping new cells into the BC atlas and estimation of the cancer subtype” at lines 638-653) and a new Suppl. Fig. 07

As suggested by the reviewer, we have now compared our mapping tool to *scmap* whose results are graphically reported below. Briefly, 75% of cells in each cell-line in the atlas were used as the training set (i.e. 26,455 cells) while the remaining 25% were used as test set (i.e. 8,821 cells). In the case of our mapping tool, the 26,455 cells of the training set were first analysed to reconstruct the breast cancer atlas from scratch. Afterwards, the 8,821 cells of the test set were mapped into the atlas as “new cells” with our mapping algorithm. Finally, the cell line type of each cell in the test set was predicted by using k-nearest neighbour classifier with k=100. In the case of *scmap* we used *n_features* parameter equal to 2,000 and k=100 neighbours for cell classification. Global performances of the two methods are reported below, and at least on our dataset, *scmap* seems not to perform very well; the performance of *scmap* (Positive Predictive Value – PPV) is worse than the one of our mapping algorithm both when considering the overall performance on all cells independently of the cell lines (panel A) and also on a cell line basis (panel B). However, we would rather not to include this comparison in the revised manuscript, as a fair comparison would require testing on multiple datasets and deeper investigation of the algorithm settings; we believe that this would go beyond the scope of this manuscript, since our aim was just to show that one of the uses of the atlas is that of mapping patients’ cells, while the mapping method could be ours but also any of those now appearing in the literature. We, however, included the quantification of the performance of our algorithm in in the revised Supplementary Figure 08A.

2.6 *When using their single-cell data to label spatial data why didn't they use existing deconvolution tools (e.g. cell2location)? Spatial data is not single-cell resolution thus each spot could be a mixture of*

profiles similar to multiple clusters in their data, deconvolution would be able to correctly quantify this mixture rather than attempting to assign the entire spot to one of the clusters.

We thank the reviewer for having pointed out an additional possible use of our single cell breast cancer atlas. We have now performed this additional analysis using cell2location to retrieve the possible cell line composition of each spot of spatial RNA-seq of the two breast cancer tumours (lines 224-230 of revised manuscript and lines 726-727 in the Methods' section). The results of cell2location tool are reported in Supplementary Figure 11 and 12 of the revised manuscript.

2.7 *Why did the authors choose to deconvolve bulk and spatial data by the cell-lines rather than by their clusters with conserved expression profiles across cell-lines?*

We thank the reviewer for the suggestion. This could have been an alternative approach, but we preferred to have cell lines as “primitives” since these are more easily interpretable from a biological point of view as they have been extensively characterised.

2.8 *For Figure 1H what was the statistical significance of the differential expression across TCGA cancer types?*

We have now added a new supplementary Table 04 with the result of the ANOVA test across the 4 TCGA cancer types for each of 22 biomarkers. We refer to this new table in the revised manuscript at line 141.

2.9 *For Figure 1I the classification task, is the accuracy coming from true positives or true negative classifications? Precision or Positive Predictive value would be a more informative statistic in this case.*

We have now modified the y-axis of Figure 1I to report the Positive Predictive Value.

2.10 *Heterogeneity: When determining whether a gene is truly heterogeneously expressed in single-cell RNAseq data it is necessary to account for the probability of detection given the Poisson sampling of sequencing data. While Supplementary Figure 8 attempt to answer this it does not take into account the heterogeneity in sequencing depth of each individual cell. The authors should calculate probability of detection in each cell of each line and then average across those for each cell-line to determine its expected detection rate:*

$$\text{prob_detect_cell_i} = \text{Poisson}(0, \text{lambda} = \text{mean_umi_per_cell_gene_X} * \text{total_umi_cell_i} / \text{mean_total_umi_per_cell})$$

We thank the reviewer for this very important suggestion and for raising this point; using the equation indicated by the reviewer, we have now computed for each cell line, the expected proportion of zeros across cells for each of the four clinical biomarkers in Figure 3A. We then tested whether the actual zero proportion was higher than the expected rate under the Poisson model, as zero inflation indicates the presence of cell heterogeneity⁸. To this end, we computed an empirical p-value for each gene and cell line, by randomly sampling from N (number of cells in the cell line) Poisson distributions using

the estimated lambdas for each cell. We thus obtained a “simulated” vector of counts, from which we computed the proportion of zero counts. This process was repeated 10,000 times to obtain an empirical distribution of the proportion of zero counts, which we then used to compute the empirical p-value. The results are reported in the revised manuscript in a new Supplementary Table 07. The heterogeneity in the expression of the clinical biomarkers is significant (p-values <0.05) for at least one of the four biomarkers in all the cell lines but two (ZR751 and BT549). Moreover, for the MDA-MB-361 cell lines, ESR1, PGR and ERBB2 are all significantly heterogeneous. We thank again the reviewer for this important observation, as some of the observed heterogeneity was indeed caused by measurement errors and not by a real biological heterogeneity. We have now included these observations in the revised manuscript (lines 272-284) and added a new paragraph in the Methods section to explain the details (new Method paragraph “Estimation of heterogeneity in biomarker expression” at lines 654-671).

2.11 *When the authors consider the cell-cycle they use a tool that classifies all cells into G1 or S/G2M, this excludes the possibility of cells having a senescent or non-cycling phenotype. Could the authors determine what portion of their cells are not cycling at all?*

We thank the reviewer for his/her comment. We classified cells according to the cell cycle from the single-cell sequencing data because of the availability of tried and tested bioinformatics approaches such as the Seurat suite, which we used for our analysis. Unfortunately, neither Seurat nor other tools are not able to distinguish between cells that are in G0 and G1 phases. To investigate cellular senescence, we used Gene Set Enrichment Analysis with a transcriptional signature of cellular senescence identified by Casella et al (6) to identify the percentage of senescence cells in MDAMB316. However, after p-value correction no cells of MDAMB361 cell-line resulted to be enriched for the expression of the senescence signature.

2.12 *Figure 3E Were these pathways the most significantly enriched? If not out of how many total pathways with FDR < 10% were they selected from? How were they selected? Figure 3D shows very little difference in proportion of G2M cells, thus it seems contradictory for cell-cycle to be the most enriched pathway differentially expressed between the two populations. Could the authors comment on why this is the case?*

In Figure 3E we reported all the significant pathways with an FDR<10% according to the Gene Set Enrichment Analysis. Regarding the apparent small difference in the proportion of G2/M cells between the HER2+ and HER2- subpopulations, we have now revised Figure 3D by splitting cells predicted in S phase from those in the G2/M phase (which we had previously merged in the original Figure 3D). We also added the p-value of the Fisher test to check for significant differences in the proportion of cells in a specific cell cycle phase between HER2+ and HER2- subpopulations. HER2- cells showed a significantly higher proportion of cells in the G2/M and S phases when compared with HER2+ cells. We have now updated Figure 3D and the relative caption in the revised manuscript.

2.13 *Drug response: In the main text it is stated the drug sensitivity was estimated for HER2 inhibitors, however the methods section explaining this analysis makes no mention of filtering for different drug mechanisms. Was the analysis considering all drugs in the database or only HER2 inhibitors? For Figure 3F is this showing direct drug potency from the CCLE database or inferred drug potency from the DREEP model?*

We have now better described in the caption of Figure 3F in the revised manuscript that the plot shows the direct measured drug potency in the CTRPv2 dataset from the CCLE (Broad Institute), which is measured as the Area Under the Curve (AUC) of the dose response measurements. In Figure 3F, we reported only HER2 inhibitors, whereas in Figure 3G, we performed the analysis on all the drugs,

however we did not explain this clearly in the original text and moreover Figure 3G was difficult to read. We have therefore simplified Figure 3G to make it more readable and we modified the main text to better explain this analysis (revised manuscript lines 332-337). Finally, we added in the Methods section a new paragraph named “Correlation between drug targets expression and drug potency” (lines 672-676) to explain how Figure 3G was derived and Supplementary Table 10 reporting the raw values used in the revised Figure 3G.

2.14 *In addition the explanation of the novel DREEP method is confusing, as this is a novel method please provide a fully detailed and clear description of how this method works. Is the section titled: "Single-cell drug sensitivity prediction" describing the DREEP method? For the validation of this tool why did the authors use a Z-score percentile - this would force cell-lines to only be recorded as sensitive to a small number of drugs even if in truth they are highly sensitive to all the tested drugs. IC50 values are standardized across lines, it would be more appropriate to choose a specific IC50 value threshold to determine if cell-lines are or are not sensitive to the drug for their gold-standard.*

Following the reviewer suggestion, in the revised manuscript we have now added a new Methods' subsection named “Description and validation of the DREEP method for single cell drug sensitivity prediction” which includes both the description of the method and its validation that in the original manuscripts were split in two different sections (revised manuscript Methods section lines 677-715). In addition, as Reviewer 1 in point 1.2 also wondered why the performance of DREEP was assessed only on 86 drugs rather than on the 450 drugs on which the algorithm was trained, we have now decided to test DREEP's performance on two different golden standards, one derived from the CTRPv2 dataset in the Cancer Cell Line Encyclopedia (CCLE) from the BROAD institute, and the other derived from Genomics of Drug Sensitivity in Cancer (GDSC) study by the Sanger Institute (2). More specifically, in the CTRPv2 dataset, drug potency of each drug in each cell line is quantified by the Area Under the Curve (AUC) of the dose response curve, but no IC50 is reported. Hence, in the original manuscript we had reported the results only for the 86 drugs (out of 450) that were also present in the GDSC dataset, which includes drug potency data measured as IC50 for about 250 small molecules (of which only 86 in common with the CTRPv2 dataset). In the revised version of the manuscript, for the “GDSC golden standard” of 86 drugs, we have now used a specific threshold for IC50 to call a cell line sensitive to a drug, as suggested by the reviewer. This IC50 thresholds had already been defined in the GDSC study, hence we used the same values as in the original publication (2). The performance of DREEP for this golden standard is in the new Supplementary Figure 18 showing both the ROC curve and the precision-recall curve.

To build the “CTRPv2 golden standard” for 450 drugs, we instead used the z-score percentiles computed from the AUC of each drug across the available cancer cell lines. We then defined a cell line sensitive to the drug if and only if its Z-score was in the lowest 5% percentile. The results of this golden standard are now shown in the revised Figure 3J. Finally, we evaluated the drug-wise performance of DREEP method for the 450 drugs, as suggested by Reviewer 1 in point 1.2, as shown in a new Supplementary Table 11. We noticed that the performance on the CTRPv2 golden standard is much better than the one using the GDSC golden standard; this is to be expected however as DREEP was trained by correlating gene expression profiles with the AUC in CTRPv2 and not with the IC50 in GDSC, and these are two independent studies that tend to agree on drug sensitivity measurements but are not perfectly overlapping.

We have now revised manuscript's main text to include a description of these new results (lines 361-371) and updated the Methods' section accordingly (lines 700-714).

Reviewer #3 (Remarks to the Author): Expert in breast cancer genomics

I read the manuscript by Gambardella and colleagues with interest. In this work, the authors profiled the transcriptome of 35,276 cells from 32 breast cancer cell lines to create a breast cancer cell line atlas. A high degree of heterogeneity was found across individual cells within the same cell line. Furthermore, single cell transcriptomic profiles from patient tumor biopsy samples were compared to the atlas and found to be heterogenous. Lastly, results from in vitro drug screening were used to predict responses to anticancer agents, providing a new resource for drug response determination.

We thank the reviewer for finding our manuscript of interest. In what follows we replied to the observations made by the reviewer.

3.1 *The interpretation of the results of this paper heavily relies on the 32 cell lines selected. More information about each cell-line should be detailed. How did the investigators ensure that the cell-lines have not been cross contaminated, and that the cluster results truly reflects tumor heterogeneity? STR profiling or other genomic analyses could be useful for this purpose.*

The reviewer is right, in our effort to be concise, we failed to properly describe and motivate our choice of cell lines. We have now expanded the Introduction section of the revised manuscript to better introduce the cell lines (revised manuscript - lines 71-77). Briefly, we chose these set of cell lines as they cover the major breast cancer tumour types and have also been extensively characterised both at the genomic and (bulk) transcriptomic level, as well as in terms of drug response thanks to publicly available large-scale screening efforts (2, 7). Following the reviewer suggestion, we have now also included the results of STR profiling on the cell lines confirming their identity, as reported in the new Suppl. Table 02 and described in the revised Method section (lines 552-554)

3.2 *For the single cell atlas generation, the authors should justify their approach of not doing batch effect correction, especially since the experiment involves two different sequencing platforms.*

We agree with the reviewer that this information was missing. We have now updated Suppl. Table 1 to include a new column indicating the batch and the sequencing platform for each of the sequenced cell lines. The main reason we did not perform batch correction is that we expected cell lines to have quite distinct transcriptional profiles, so that the effect of batch as compared to the biological variability should be minimal. Usually, batch correction is required if one needs to compare the same cell type or tissue across different patients distributed in multiple batches, where the batch effect could be much higher than biological effect or biological variability. To address the reviewer's concern, we have now performed a Principal Component Analysis (PCA) of all the single cell data as now reported in revised Supplementary Figure 01B. To this end, we first converted single cell expression profiles within a cell line to a pseudo-bulk expression profile by computing the average expression of each gene across the single cells. In the PCA plot each cell line is then represented by a dot, whose colour represents the batch (i.e. the cell lines sequenced in the same flow cell) and whose shape, the sequencing platform. As it can be appreciated in revised Supplementary Figure 01B, no major batch effect was observed, hence we decided not to correct for batch effect, since batch correction methods may mask biological variability (8). We have now included these observations in the revised manuscript in the Methods section (lines 594-598).

3.3 *Line 87, for Fig 1D, the authors should also include the relative expression of these genes, instead of just showing the percentage of cells expressing them, to better clarify the phrase "highly expressed".*

The reviewer is right. Following the reviewer suggestions, we have now added a Supplementary Figure 03, analogous to Figure 1D, where we show the relative expression of the genes. The reason we used the term “highly expressed” when referring to the percentage of cells expressing a gene as is that this measure is strongly correlated to the average expression across single cells cell (that is the higher the percentage of cells expressing a gene, the more its average expression, see please also response to point 1.1 and Supplementary Figure 16B).

3.4 *Line 88-89, the authors should quantify the similarity between triple-negative BC cell lines with basal-like cell to support their statement, in addition to listing the common genes expressed.*

Following the reviewer, suggestion we have now selected a set of genes known to be specifically expressed in basal epithelial cells (9–15) and used this set, in conjunction with Gene Set Enrichment Analysis (GSEA), to quantify the extent of expression of these genes and its significance in triple-negative BC cell line. The results of this analysis are reported in the new Suppl. Table 03 and show that 9 out 15 triple-negative cell lines significantly (FDR<0.05) express the basal biomarkers. We have modified the manuscript to describe this additional analysis in the main text (line 107) and added a new subsection in the Methods (lines 608-615).

3.5 *Lines 125-127, it is hard to distinguish LumA, LumB, and maybe Her2 from just the 20 biomarkers being derived. The findings shown in Supplementary Figure 4 are not adequate to support the statement made.*

We have now removed the original Supplementary Figure 4 and the statement regarding the possible prognostic value of the genes, following the suggestion of the reviewer. Regarding the value of the 20 biomarkers, we received a somewhat similar observation from Reviewer 2 (in point 2.2). During revision, we realised that we had wrongly excluded two of the biomarkers (MAGEA4 and XAGE2) out of the original 22 biomarkers (one per cluster), which we have now included in this revised version of the manuscript. The reason why MAGEA4 and XAGE2 were not mapped to TCGA dataset in the original manuscript was the way we pre-processed TGCA dataset. Specifically, during the pre-processing step, we had removed poorly expressed genes, i.e. those genes whose average expression across the 937 breast cancer patients from the TCGA collection was in the lower 5% percentile, and MAGE4 and XAGE2 were among these ones. However, we noticed that in a few patients these genes were indeed highly expressed. Hence, we have now removed this superfluous pre-processing step and thus recovered MAGE4 and XAGE2. We have now consequently updated Figure 1H and Figure 1I in the revised manuscript showing all the 22 biomarker genes from our BC atlas. In the updated Figure 1H, MAGE4 and XAGE2 are highly expressed only in a subset of triple-negative breast cancer patients and in a subset of HER2+ /ER negative patients; interestingly, MAGE4 has been previously reported in the literature as overexpressed in HER2+ /ER negative patients using proteomic profiling (5). Regarding the observation of the reviewer that from the 20 biomarkers (now 22) it is not possible to readily distinguish the subtypes, we would like to point out that in the revised Figure 1I, we show that it is indeed possible to classify patients in LumA and LumB subtypes with this signature, although the classic PAM50 does a better job, as expected. However, the PAM50 signature was specifically designed to classify BC subtypes, whereas the signature we derived from the single cell atlas was automatically generated by clustering single cell data without using any a priori information on the BC subtypes. The reason we compared our signature to the PAM50 was just to show that clustering of single-cell data can be used to automatically detect biomarker genes that can be clinically relevant. Please also refer to our reply to point 3.8.

3.6 Lines 127 - *The authors cited that the expression of KLK10 in a TNBC subset represents an opportunity for biomarker discovery. However, neither tamoxifen nor trastuzumab are used in TNBC, and thus the relevance of KLK10 in TNBC is questionable*

We agree with the reviewer that this observation is not clinically relevant, and we have now removed this statement from the revised manuscript.

3.7 Lines 131 - *The presence of ACTG2 in a subset of TNBC and its relationship to cell proliferation and platinum-based chemosensitivity is highlighted here. The authors should provide evidence for this statement here using the cell-lines available e.g. HS578T/MX1.*

Our intention here was just to highlight the relevant literature for some of the biomarkers that we had automatically identified using our single cell dataset, and we thought that this previously published relationship between ACTG2 and platinum-based chemosensitivity was interesting to mention. To comply the reviewer request, we have now performed experiments in these two cell lines, as they show considerably higher expression of ACTG2 than all other cells in the dataset (new Supplementary Figure 06A,B). We treated both cell lines with cis-platin and derived a dose response curve by measuring cell viability at 72 hours (new Supplementary Figure 06C). The results of these experiments confirm sensitivity to cisplatin of the HS578T cell line and to a lesser extent of the MX1 cell line. These results are described in the main text (lines 150-156).

3.8 Lines 133 - *The performance of the 20 genes against the PAM50 gene signature was deemed to be "overall comparable". This statement should be backed up by appropriate statistical tests.*

We agree with the reviewer that our statement was qualitative and not supported by statistical analysis. Following the reviewer's suggestion, we have now performed a thorough statistical analysis using a cross-validation technique. Specifically, we divided the set of 937 patients from TGCA, for whom cancer subtype was annotated, into training set of 625 patients (two thirds of the patients) and test set of 312 patients (one third of the patients). The training set was used to train a standard classifier algorithm (XGBoost) with the chosen gene signature (PAM50, scCCL or scCCL+HER2) while the test set was used to compute the classification accuracy (the percentage of patients correctly classified) for each tumour subtype. We repeated this process 3 times (i.e. 3-fold cross validation), each time randomly assigning patients to the training set and to the test set and then computing the classification accuracy. We also computed the probability of correctly classifying a patient when making a random guess, this simply corresponds to the percentage of patients in the dataset for each of the tumour subtypes. In the revised Figure 1I, we have now reported the average value of the accuracy for each tumour subtype and its standard deviation for the three signatures, together with p-values following a statistical t-test between PAM50 and each one of the other two signatures. We also indicated with a red line the accuracy obtained by a random guess. The statistical test shows that all the three signatures can classify patients better than a random guess; however, whereas for the basal subtype PAM50 is indistinguishable from the other two signatures, PAM50 is indeed better in classifying the other three subtypes. We would like to point out, however, that the reason for performing the comparison with the PAM50 signature in the original manuscript was not to suggest to use our signature in place of the PAM50, but rather to validate the atlas-derived signature. Specifically, our signature is automatically derived by clustering single cell data and, unlike the PAM50, does not use any prior knowledge on the breast cancer subtypes. Nevertheless, it can distinguish among the current clinical subtypes better than random, thus suggesting that the 22 biomarker genes in our signature are clinically relevant, despite being different from the 50 genes in the PAM50. Our approach could be very useful for automatically identifying signature for less studied tumours for which no signature is currently available, and no clear clinical subtypes have been

identified. Considering these results, we have now modified the main text by removing the statement “overall comparable” (revised manuscript lines 157-166), added a new paragraph in the revised Methods section (lines 716-725) and updated Figure 1I.

3.9 *The authors derived a methodology to assign cell line data to patient tumor profiles, starting with single cell profiles of 5 TNBC cases. Curiously 1 case - TNBC5 was mapped as "luminal" instead. No further information on the 5 TNBC samples were provided, including how the original diagnosis was made. Similarly, spatial transcriptomics of 2 cases were used to test the algorithm, picking up HER2-overexpressing or triple negative cell populations. Corresponding IHC and HER2 FISH data should be demonstrated to demonstrate accuracy of the original diagnoses.*

The data for the 5 TNBS cases were obtained from a recently published work in Nature Biotechnology from the lab of Prof. Nicholas Navin at MD Anderson. The authors wrote in that manuscript that they performed classification of cancer subtypes “by pathological evaluation of hematoxylin and eosin-stained tissue sections, immunohistochemistry analysis of estrogen receptor (<1%) and progesterone receptor (<1%) and fluorescence in situ hybridization analysis of HER2 amplification using the CEP-17 centromere control probe (ratio of HER2 to CEP-17 < 2.2).” I contacted Prof Navin who confirmed that patients were enrolled in the ARTEMIS clinical trial at MD Anderson for neoadjuvant chemotherapy treatment, and these were core biopsy samples collected prior to any treatment, but he could not share any information as the clinical trial is still ongoing. We ourselves were surprised and intrigued by the results for TNBC5 and thus we have now further investigated why our algorithm mapped the majority of these cells as luminal. We observed that TNBC5 was the only patient out of the five patients to highly express the androgen receptor AR and the transcription factor FOXA1 (refer to revised Suppl. Fig. 09); interestingly, co-expression of AR and FOXA1 has been reported in about 15% of triple-negative breast, and it is considered a distinct class of basal-like tumour inducing an estrogen-like gene signature, which may explain our result (16). Moreover, out of the Triple Negative cell lines included in our atlas, none expresses both AR and FOXA1 (refer to Figure 1D), hence our mapping tool may have assigned patient TNBC5 to luminal cell lines as it found these cell lines to be more similar than any of the triple negative cell lines. Finally, we applied the PAM50 signature to the pseudo-bulk expression profiles of the five TNBC patients. Pseudo-bulk refers to the use of single cell expression profiles to compute the average gene expression and thus simulate a bulk gene expression measurement. The results of the PAM50 classification, now in revised Suppl. Table 06, show that whereas patients TNBC1 to 4 were correctly classified as basal-like with about 99% probability, TNBC5 had a probability of just 4% of being basal-like, 47% of being HER2-enriched, and 48% of being luminal (A or B), again indicating the peculiarity of this patient. We have now added these observations in the revised manuscript (lines 190-205) and in revised Supplementary Figure 09 and Supplementary Table 6.

Finally regarding the two spatial transcriptomics datasets, we did not perform these experiments ourselves, but they were publicly available from the 10X Genomics company website (<https://wp.10xgenomics.com/resources/datasets>), which obtained the original tissue from the BIOIVT Astarand company with a board-certified pathologist review, whose results however are not available to download from the 10X Genomics website. We agree that this is suboptimal hence we have now highlighted in the manuscript (lines 216-217) this limitation to make the reader aware of it.

3.10 *The assignment of specific tumor cell lines to the composition of whole tumor samples (using bulk transcriptomic data), while interesting, needs further validation with corresponding single cell data from the same tumor samples.*

We agree with the reviewer, but we would like to point out that the bioinformatic deconvolution algorithm, which we used to assign tumour cell lines from the bulk profile, was not developed by us (it was cited as reference 49 in the original manuscript). This tool is named “Bisque” and it was published in Nature Communication in 2020 where its performance was extensively validated. We recognise now that the way we wrote this paragraph did not clearly convey this important information. In our manuscript, we just meant to show that our single-cell atlas can be used in combination with deconvolution tools developed by other groups, such as Bisque, to assign a cell line based composition to a bulk tumour profile. This is the reason why we did not perform a full validation of Bisque in our original manuscript. However, to comply with the reviewer request, we have now performed a validation of Bisque when applied to our single cell dataset. We agree that the ideal case would have been to obtain both bulk and single cell transcriptomics from the same tumour samples, however this is difficult as it requires a large biopsy, which is rarely available at least for breast cancer. Moreover it would require collection of biopsies of suitable size and generation of additional data that would go beyond the scope of this current work. We also searched the literature for suitable datasets that we could use, but we could not find any. We thus decided to use bulk transcriptomic data of breast cancer cell lines that are publicly available from the Cancer Cell Line Encyclopedia of the BROAD Institute and whose single-cell profiles are present in our atlas; we could find a total of 29 bulk transcriptomic profiles out of 32 cell lines in our atlas. We then applied the Bisque tool to predict from each of the 29 bulk gene expression profiles its cell line composition using the single cell atlas; if the Bisque tool were to work perfectly, then we would expect Bisque to predict each bulk profile to be 100% composed of the corresponding cell line in the atlas, whereas if it did not work at all, just a small fraction would be assigned to the corresponding cell line. In the revised Supplementary Figure 13, we have now reported the results of Bisque. For each of the 29 bulk gene expression profiles, Bisque correctly predicted that the largest fraction of cells composing it came from the corresponding cell line in the atlas with a range between 40% to 80%. We have now added a new paragraph in the main text (lines 231-242) and revised Suppl. Fig. 13 to present these new results.

3.11 *Fig 1H, the authors should do a statistical test between PAM50 and scCCL markers. If scCCL markers are not doing better when compared to PAM50, then why should they be used?*

As we described in our reply to point 3.8, we have now revised Figure 1I and added the proper statistical tests to compare PAM50 and scCCL biomarkers’ performance. The reviewer asks why one should use the scCCL markers in place of the PAM50. Indeed, one should not. Our aim was not the generation of a novel gene signature to classify BC patients better than PAM50, but rather to validate the single cell atlas by showing it can be used to automatically detect genes that are relevant for the disease. The 22 scCCL marker genes were chosen by selecting one gene for each of the 22 clusters; for each cluster; the selected gene was the one most differentially expressed between cells in that cluster and all the remaining cells. On the contrary, the PAM50 genes were carefully and manually selected to maximise classification of the different breast cancer subtypes, whereas single cell sequencing based biomarkers are fully automatic and can be easily translated to other cancer types.

3.12 *Line 183 - "JIMT-1 is an HER2-overexpressing cell line with an amplified 184 ERBB2 locus, but no HER2+ patient was mapped to this cell line" Figure 2E however seems to show that JIMT-1 maps to HER2+ patient TCGA-E2-A1B0*

We thank the reviewer for pointing this out and we apologise for the lack of clarity that caused this misunderstanding. The statement in line 183 of the original manuscript referred to the results in Figure 2F reporting the percentage of TCGA patients mapped to each cell line, and where no HER2+ patient mapped to the JIM1T cell line. Indeed, this seems to contradict results in Figure 2E where patient

TCGA-E2-A1B0 is mapped to the JIM1T cell line. The explanation of this apparent contradiction stems from the way Figure 2F was constructed: the cell line composition of each TCGA patient is estimated from bulk gene expression profile by means of the deconvolution tool “Bisque” (for more details please refer to our reply to point 3.10); since each patient is usually predicted by Bisque to be composed by more than one cell line, in order to assign a single cell line to the patient, we chose the cell line making up the largest fraction in the patient’s cell line composition. In the case of patients TCGA-E2-A1B0 in Fig.2E, the cell line with the largest fraction would be HCC1954. In the light of this explanation, we have now added more details to better explain how Figure 2F was obtained (revised manuscript lines 247-250 and Figure 2 caption) .

3.13 *For the cancer subtype classification section, the authors should show how frequent their algorithm will be able to predict tumor subtype correctly (may be using simulated data), given that the data from scRNAseq, spatial sequencing, and bulk RNAseq have showed some inaccurate classification to certain degree.*

To address the reviewer concern we have now performed a quantification of the classification performance both of our single cell mapping algorithm from scRNA-seq, and of Bisque from bulk RNA-seq in terms of how many correct predictions are made out of the total predictions (Positive Predictive Value). We first tested our single cell mapping algorithm’s performance in correctly classifying the tumour subtype from scRNA-seq; to this end, we divided single cell transcriptional profiles in the atlas in a training set and a test set. The training set included 75% of the cells in each cell line, for a total of 26,455 cells, whereas the training set consisted of the remaining 25% of cells in each cell line, for a total of 8,821 cells. The mapping algorithm was trained on the training set and tested on the test set to quantify its performance in correctly assigning each single cell to the correct cell line, and hence tumour subtype. As shown in the revised Supplementary Figure 08B, the accuracy of the algorithm was about 90%. We then assessed the performance of the *Bisque* deconvolution algorithm in classification of TNBC patients, when using Bisque to predict cell line composition, and then by assigning to each TNBC patient, the subtype of the cell line making up the largest fraction in the predicted cell line composition. These new results are reported in new Figure 2G, and described the revised main text (lines 254-259)

3.14 *Line 202-204, the claim the authors made based on Supplementary Figure 8 require additional justifications / data, given that the p-value is so high.*

We apologise with the reviewer for not having been clearer in explaining our reasoning. In the original Supplementary Figure 8 , we did expect p-values to be high, that is not significant, as this proves a lack of any significant correlation between the number of cells expressing a receptor and the sequencing depth. Indeed, one of the problems of single cell sequencing is the low capture efficiency of RNA molecules coupled to the limited number of sequences per cell (aka reads) that can lead to lack of measurement of an mRNA in a cell (*i.e.*, a zero count) not because the cell is not expressing it but simply because of a technical measurement limitation. To exclude this source of error in Suppl. Fig. 8 we correlated the number of uniquely mapped reads (UMI) of a cell line to the percentage of cells in the cell line expressing each one of the four biomarkers; had these quantities been significantly correlated, then this would have implied that differences in the number of reads can explain how many cells in a cell line express the receptor and thus point to an apparent heterogeneity which is not biological but caused by measurement artefacts of the single-cell sequencing technology. In addition to this, we have now performed new analyses as suggested by Reviewer 2 in point 2.10. Specifically, Reviewer 2 pointed out that to call a gene truly heterogeneously expressed in single-cell RNAseq data it is necessary to account for the probability of detection of that gene by assuming a Poisson sampling of sequencing data. Hence, using the formula indicated by Reviewer 2 in point 2.10, we have now computed for each cell line, the expected proportion of zero counts across cells for each one of the four

clinical biomarkers. We then tested for each biomarker gene whether the actual proportion of cells with a zero count was higher than the expected one under the Poisson model, as zero “inflation” indicates the presence of a significant cell heterogeneity not caused by measurement artefacts (17). To this end, we computed an empirical p-value for each gene and cell line, by randomly sampling from N (number of cells) Poisson distributions. We thus obtained a “simulated” vector of counts, from which we computed the proportion of zero counts. This process was repeated 10,000 times to obtain an empirical distribution of the proportion of zero counts, which we then used to compute the empirical p-value. The results are reported in the revised manuscript in a new Supplementary Table 07. The heterogeneity in the expression of the clinical biomarkers was found to be significant (p-values <0.05) for at least one of the four biomarkers in all of the cell lines but two (ZR751 and BT549). Moreover, for the MDA-MB-361 cell lines, three biomarkers (ESR1, PGR and ERBB2) are significantly heterogeneous. We have now included these observations in the revised manuscript (lines 272-284) and added a new paragraph in the Methods section to explain the details (new Method paragraph “Estimation of heterogeneity in biomarker expression” at lines 654-671).

3.15 *At the last paragraph of Discussion, the authors had heavily emphasized how important the single-cell RNAseq is, but throughout the manuscript, there wasn't much details/analysis on getting the differential expression within the same tumor, instead most of the tumor subtypes were clustered according to their own origin (triple negative), or they were grouped together (luminal).*

We allowed ourselves to be a bit speculative in the discussion section. Our manuscript is focussed on single cell transcriptomics of cancer cell lines with the aim of releasing a comprehensive and unique dataset for the research community and to explore the relevance of single cell technology in diagnosis and drug treatment selection. We agree that the direct clinical value of single cell sequencing is still an open question. However, we believe that our manuscript lays some of the foundational work towards this aim. Using the single cell atlas as a framework, future work focusing on single cell transcriptomics of patients' biopsies will be needed to prove the relevance of single cell atlas of cancer cell lines to map single cell sequencing from tumour biopsies in the clinical setting, however this is a substantial effort and outside the scope of the current manuscript.

3.16 *Overall, while the concept of using cell-line information as a base to infer heterogeneity in tumor samples is interesting, further validation is required. Ideally, a good validation set of cases should have both bulk transcriptomic and single cell gene expression data. Spatial transcriptomic data will be useful as well. Most crucially, the cases should be properly annotated histopathologically.*

We thank the reviewer for finding the idea of cell-line composition of a tumour interesting; and we do agree that this idea requires further validation; however a proper validation would require extensive time and funding to set up a whole new project including collection of biopsies of suitable size to perform both bulk and single cell sequencing, generation of additional data and a proper comparison with histopathology for a large number of samples; however such an effort is well beyond the scope of this current work, indeed we are currently looking for funding for this purpose, hence the importance of this manuscript. We believe however that despite this limitation, this manuscript deserves publication as cell line based composition was only one of its aspects, indeed the main objective was to make available to the community this unique resource, i.e. the single cell atlas, and to show how single cell sequencing can be used not only to detect immune cell infiltration in a tumour biopsy as currently done, but also in cancer cell lines to identify clinically relevant biomarkers, to detect heterogeneity in expression of drug targets, and to predict drug sensitivity. We also think that this manuscript will help to make the case for the usefulness of sequencing all the cancer cell lines currently available (more than 1000 cell lines across several tumour types) at the single cell level, which can only be done by large

scale consortium or large genome centres, and to spur further research in the use of cell lines as primitives to study tumour heterogeneity, and lastly in the potential clinical applications of single cell sequencing.

Minor comments:

1) In the Results section, the authors described luminal BC cells and TNBC cells forming "islands", "peninsulas" and "archipelagos". This is hard to visualize and it is unclear what the authors are trying to elucidate from this description. Perhaps figure annotations can be used to highlight this.

2) Figure 1F/Supplementary Figure 2 - colors used should be defined

3) Color definitions of LA, LB, H, TNA, TNB are sometimes different across figures. This makes interpretation of figures difficult.

4) Line 163, it should be Supplementary Figure 6B,C (instead of Supplementary Figure 6C,D)

5) Line 165, it should be Supplementary Figure 6B (instead of Supplementary Figure 6C)

6) Line 181, "With" instead of "We" this approach

We thank the reviewer for these observations, we have now modified the text and the figures accordingly.

References

1. M. G. Rees *et al.*, Correlating chemical sensitivity and basal gene expression reveals mechanism of action. *Nat Chem Biol.* **12**, 109–116 (2016).
2. F. Iorio *et al.*, A Landscape of Pharmacogenomic Interactions in Cancer. *Cell.* **166**, 740–754 (2016).
3. X.-F. Le *et al.*, Genes Affecting the Cell Cycle, Growth, Maintenance, and Drug Sensitivity Are Preferentially Regulated by Anti-HER2 Antibody through Phosphatidylinositol 3-Kinase-AKT Signaling*. *J. Biol. Chem.* **280**, 2092–2104 (2005).
4. J. M. Henwood, R. N. Brogden, Etoposide. *Drugs.* **39**, 438–490 (1990).
5. T. Cabezón *et al.*, Proteomic Profiling of Triple-negative Breast Carcinomas in Combination With a Three-tier Orthogonal Technology Approach Identifies Mage-A4 as Potential Therapeutic Target in Estrogen Receptor Negative Breast Cancer*. *Mol. Cell. Proteomics.* **12**, 381–394 (2013).
6. G. Casella *et al.*, Transcriptome signature of cellular senescence. *Nucleic Acids Res.* **47**, 7294–7305 (2019).
7. M. G. Rees *et al.*, Correlating chemical sensitivity and basal gene expression reveals mechanism of action. *Nat. Chem. Biol.* **12**, 1–10 (2015).
8. M. Büttner, Z. Miao, F. A. Wolf, S. A. Teichmann, F. J. Theis, A test metric for assessing single-cell RNA-seq batch correction. *Nat. Methods.* **16**, 43–49 (2019).
9. X. Dai, H. Cheng, Z. Bai, J. Li, Breast cancer cell line classification and Its relevance with breast tumor subtyping. *J. Cancer.* **8**, 3131–3141 (2017).
10. A. M. Sawayama, H. Tanaka, T. J. Wandless, Total Synthesis of Ustiloxin D and Considerations on the Origin of Selectivity of the Asymmetric Allylic Alkylation. *J. Org. Chem.* **69**, 8810–8820 (2004).
11. R. M. Neve *et al.*, A collection of breast cancer cell lines for the study of functionally distinct cancer subtypes. *Cancer Cell.* **10**, 515–527 (2006).
12. C. M. Perou *et al.*, Molecular portraits of human breast tumours. *Nature.* **406**, 747–752 (2000).
13. F. Bertucci *et al.*, *Cancer Res.*, in press, doi:10.1158/0008-5472.CAN-04-2696.
14. S. Hashmi Dairkee, B. Mayall, H. Smith, A. Hackett, MONOCLONAL MARKER THAT PREDICTS EARLY RECURRENCE OF BREAST CANCER. *Lancet.* **329**, 514 (1987).
15. M. Riaz *et al.*, miRNA expression profiling of 51 human breast cancer cell lines reveals subtype and driver mutation-specific miRNAs. *Breast Cancer Res.* **15**, R33 (2013).
16. S. Guiu *et al.*, Prognostic value of androgen receptor and FOXA1 co-expression in non-metastatic triple negative breast cancer and correlation with other biomarkers. *Br. J. Cancer.* **119**, 76–79 (2018).
17. T. H. Kim, X. Zhou, M. Chen, Demystifying "drop-outs" in single-cell UMI data. *Genome Biol.* **21**, 196 (2020).

REVIEWERS' COMMENTS

Reviewer #1 (Remarks to the Author):

The authors have answered all my questions.

Bence Szalai

Reviewer #2 (Remarks to the Author):

I thank the authors for addressing all of my questions and concerns. I believe the manuscript is much improved and I have no further concerns.

Reviewer #3 (Remarks to the Author):

The Authors have addressed our concerns adequately